# Human axial progenitors generate trunk neural crest cells in vitro

Thomas JR Frith[1], Ilaria Granata[2], Matthew Wind[1], Erin Stout[1], Oliver Thompson[1], Katrin Neumann[3], Dylan Stavish[1], Paul R Heath[4], Daniel Ortmann[5], James OS Hackland[1], Konstantinos Anastassiadis[3], Mina Gouti[6], James Briscoe[7], Valerie Wilson[8], Stuart L Johnson[9], Marysia Placzek[9,10], Mario R Guarracino[2], Peter W Andrews[1], Anestis Tsakiridis[1,10]*

[1]Centre for Stem Cell Biology, Department of Biomedical Science, The University of Sheffield, Sheffield, United Kingdom; [2]Computational and Data Science Laboratory, High Performance Computing and Networking Institute, National Research Council of Italy, Napoli, Italy; [3]Stem Cell Engineering, Biotechnology Center, Technische Universität Dresden, Dresden, Germany; [4]Sheffield Institute for Translational Neuroscience, University of Sheffield, Sheffield, United Kingdom; [5]Anne McLaren Laboratory, Wellcome Trust-MRC Stem Cell Institute, University of Cambridge, Cambridge, United Kingdom; [6]Max Delbrück Center for Molecular Medicine, Berlin, Germany; [7]The Francis Crick Institute, London, United Kingdom; [8]MRC Centre for Regenerative Medicine, Institute for Stem Cell Research, School of Biological Sciences, University of Edinburgh, Edinburgh, United Kingdom; [9]Department of Biomedical Science, University of Sheffield, Sheffield, United Kingdom; [10]The Bateson Centre, University of Sheffield, Sheffield, United Kingdom

*For correspondence:
a.tsakiridis@sheffield.ac.uk

Competing interests: The authors declare that no competing interests exist.

**Abstract** The neural crest (NC) is a multipotent embryonic cell population that generates distinct cell types in an axial position-dependent manner. The production of NC cells from human pluripotent stem cells (hPSCs) is a valuable approach to study human NC biology. However, the origin of human trunk NC remains undefined and current in vitro differentiation strategies induce only a modest yield of trunk NC cells. Here we show that hPSC-derived axial progenitors, the posteriorly-located drivers of embryonic axis elongation, give rise to trunk NC cells and their derivatives. Moreover, we define the molecular signatures associated with the emergence of human NC cells of distinct axial identities in vitro. Collectively, our findings indicate that there are two routes toward a human post-cranial NC state: the birth of cardiac and vagal NC is facilitated by retinoic acid-induced posteriorisation of an anterior precursor whereas trunk NC arises within a pool of posterior axial progenitors.
DOI: https://doi.org/10.7554/eLife.35786.001

## Introduction

The neural crest (NC) is a multipotent cell population which arises in the dorsal neural plate/non-neural ectoderm border region during vertebrate embryogenesis. Studies utilising chick and amphibian embryos have indicated that different levels of BMP, WNT and FGF signals, emanating from the mesoderm/non-neural ectoderm, orchestrate NC induction and specification (*Stuhlmiller and García-Castro, 2012*). This occurs via the action, first of neural plate border-specific transcription factors such as PAX3/7, MSX and ZIC family members, and then via definitive NC-specifiers (e.g. SOX9/10) (*Simões-Costa and Bronner, 2015*). Once specified, NC cells undergo epithelial-to-mesenchymal transition (EMT), exit the neural tube, and migrate to generate various cell types. The identity of NC

products correlates with their position along the anteroposterior (A-P) axis, which is in turn reflected by the expression of HOX gene paralogous groups (PGs). Cranial NC cells give rise to mesoectodermal derivatives (e.g. dermis, cartilage, bone), melanocytes, neurons and glia colonizing the head (*Le Douarin et al., 2004*) and are divided into an anterior HOX-negative and a posterior HOX PG (1-3 )-positive domain. The latter also includes cells contributing to heart structures (termed cardiac NC) (*Le Douarin et al., 2004*; *Kirby et al., 1983*). Vagal NC cells, which are located between somites 1–7, are marked by the expression of HOX PG(3-5) members (*Kam and Lui, 2015*; *Fu et al., 2003*; *Chan et al., 2005*) and generate the enteric nervous system (ENS) (*Le Douarin and Teillet, 1973*). HOX PG(5-9) -positive NC cells at the trunk level (*Kam and Lui, 2015*; *Nguyen et al., 2009*; *Ishikawa and Ito, 2009*; *Huber et al., 2012*) produce sympathoadrenal cells, which in turn give rise to sympathetic neurons, neuroendocrine cells, and melanocytes (*Le Douarin and Teillet, 1974*).

An attractive approach for studying human NC biology and modelling NC-associated developmental disorders (neurocristopathies) involves the in vitro differentiation of human pluripotent stem cells (hPSCs) toward NC cells. Conventional protocols to obtain NC from hPSCs are based on the production of a neurectodermal intermediate, via TGFβ signalling inhibition, which is subsequently steered toward a NC fate, usually through stimulation of WNT activity combined with the appropriate levels of BMP signalling (*Lee et al., 2007*; *Menendez et al., 2011*; *Chambers et al., 2012*; *Hackland et al., 2017*). These strategies yield NC cells of an anterior cranial character lacking HOX gene expression and the generation of more posterior HOX+ NC subtypes typically relies on the addition of retinoic acid (RA) and/or further WNT signalling stimulation (*Huang et al., 2016*; *Oh et al., 2016*; *Fattahi et al., 2016*; *Denham et al., 2015*). However, these signals fail to efficiently induce a high number of NC cells of a HOX PG(5-9) +trunk identity from an anterior cranial progenitor. Therefore, the generation of trunk NC derivatives such as sympathoadrenal cells often requires the flow cytometry-based purification of small cell populations positive for lineage-specific fluorescent reporter (*Oh et al., 2016*) or cell surface markers (*Abu-Bonsrah et al., 2018*), a time-consuming and laborious approach.

A number of studies in chicken and mouse embryos employing both fate mapping and lineage tracing have shown the existence of a posterior NC progenitor entity, which is distinct from its more anterior counterparts and potentially co-localises with a pool of caudally-located axial progenitors (*Catala et al., 1995*; *Albors et al., 2016*; *Javali et al., 2017*; *Schoenwolf et al., 1985*; *Schoenwolf and Nichols, 1984*; *Wymeersch et al., 2016*). These progenitors include a bipotent stem cell-like population that fuels embryonic axis elongation through the coordinated production of spinal cord neurectoderm and paraxial mesoderm (PXM) (*Tzouanacou et al., 2009*) (reviewed in (*Steventon and Martinez Arias, 2017*) and (*Henrique et al., 2015*). In both mouse and chick embryos these neuromesodermal progenitors (NMPs) are located in the node/streak border and the caudal lateral epiblast during early somitogenesis, and later in the chordoneural hinge within the tailbud (TB) (*Wymeersch et al., 2016*; *Cambray and Wilson, 2007*; *Cambray and Wilson, 2002*; *Brown and Storey, 2000*; *McGrew et al., 2008*). No unique NMP markers have been determined to date and thus, molecularly, NMPs are defined by the co-expression of the pro-mesodermal transcription factor Brachyury (T) and neural regulator SOX2 (*Tsakiridis et al., 2014*; *Olivera-Martinez et al., 2012*; ). Furthermore, they express transcripts which are also present in the primitive streak (PS) and TB, marking committed PXM and posterior neurectodermal progenitors such as *Cdx* and *Hox* gene family members, *Tbx6* and *Nkx1-2* (*Albors et al., 2016*; *Javali et al., 2017*; *Cambray and Wilson, 2007*; *Gouti et al., 2017*; *Amin et al., 2016*). T and SOX2 have a critical role, in conjunction with CDX and HOX proteins, in regulating the balance between NMP maintenance and differentiation by integrating inputs predominantly from the WNT and FGF signalling pathways (*Wymeersch et al., 2016*; *Gouti et al., 2017*; *Amin et al., 2016*; *Young et al., 2009*; *Koch et al., 2017*). The pivotal role of these pathways has been further demonstrated by recent studies showing that their combined stimulation results in the robust induction of T + SOX2+ NMP like cells from mouse and human PSCs (*Turner et al., 2014*; *Lippmann et al., 2015*; *Gouti et al., 2014*).

NMPs/axial progenitors appear to be closely related to trunk NC precursors in vivo. Specifically, trunk NC production has been shown to be controlled by transcription factors which also regulate cell fate decisions in axial progenitors such as CDX proteins (*Sanchez-Ferras et al., 2012*; *Sanchez-Ferras et al., 2014*; *Sanchez-Ferras et al., 2016*) and NKX1-2 (*Sasai et al., 2014*). The close relationship between bipotent axial and posterior NC progenitors is further supported by fate mapping experiments involving the grafting of a portion of E8.5 mouse caudal lateral epiblast T+SOX2+ cells

(*Wymeersch et al., 2016*) and avian embryonic TB regions (*Catala et al., 1995*; *McGrew et al., 2008*) which have revealed the presence of localised cell populations exhibiting simultaneously mesodermal, neural and NC differentiation potential. Furthermore, retrospective clonal analysis in mouse embryos has shown that some posterior NC cells originate from progenitors which also generate PXM and spinal cord neurectoderm (*Tzouanacou et al., 2009*). This finding is in line with lineage tracing experiments employing NMP markers such as *T* (*Anderson et al., 2013*; *Feller et al., 2008*; *Garriock et al., 2015*; *Perantoni et al., 2005*), *Nkx1-2* (*Albors et al., 2016*), *Foxb1* (*Turner et al., 2014*; *Zhao et al., 2007*) and *Tbx6* (*Javali et al., 2017*) as Cre drivers showing that axial progenitor descendants include NC cells at caudal levels. Together these findings suggest that the trunk/lumbar NC is likely to originate from a subset of axial progenitors arising near the PS/TB.

Here we sought to determine whether trunk NC is also closely related to NMPs in the human and thus define a robust and improved protocol for the production of trunk NC cells and their derivatives from hPSCs. We show that hPSC-derived, 'pre-neural' axial progenitors contain a subpopulation that displays a mixed early NC/NMP transcriptional signature and thus is likely to represent the earliest trunk NC precursors. We demonstrate that T+ neuromesodermal potent axial progenitor cultures are competent to efficiently generate trunk NC cells, marked by thoracic HOX gene expression. This transition to trunk NC appears to take place via the maintenance of a CDX2/posterior HOX-positive state and the progressive amplification of an NC gene regulatory network. We also show that 'caudalisation' via RA treatment of anterior NC precursors leads to the acquisition of a mixed cardiac/vagal NC identity rather than a trunk NC character and define novel markers of distinct posterior NC subtypes. Finally, we utilise our findings to establish a protocol for the in vitro generation of PHOX2B+ sympathoadrenal cells and sympathetic neurons at high efficiency from cultures of posterior axial progenitor-derived trunk NC cells without the need for FACS-sorting to select for minor precursor subpopulations. Taken together these findings provide insight into the mechanisms underpinning the 'birth' of human NC cells at different axial levels and pave the way for the in vitro modelling of trunk neurocristopathies such as neuroblastoma.

## Results

### Transcriptome analysis of human axial progenitors

We and others have previously shown that combined stimulation of the WNT and FGF signalling pathways in PSCs leads to the production of a high (>80%) percentage of T+SOX2+ cells. The resulting cultures resemble embryonic posterior axial progenitors, including NMPs, both in terms of marker expression and developmental potential (*Gouti et al., 2017*; *Turner et al., 2014*; *Lippmann et al., 2015*; *Gouti et al., 2014*; *Tsakiridis and Wilson, 2015*). To interrogate the transcriptome changes associated with the induction of such progenitors in a human system and identify the presence of trunk NC precursors, we carried out RNA sequencing (RNAseq) following 3- day treatment of hPSCs with recombinant FGF2 and the WNT agonist/GSK-3 inhibitor CHIR99021 (CHIR). As reported previously, most cells emerging under these conditions co-expressed T and SOX2 as well as CDX2 (*Figure 1A*, *Figure 2—figure supplement 1B*). We found that the transcriptomes of axial progenitors/NMPs and hPSCs were distinct from each other (*Figure 1—figure supplement 1A,B*) with marked global gene expression changes accompanying the acquisition of an axial progenitor character: 1911 and 1895 genes were significantly (padj <0.05; Fold Change $\geq$ 2) up- and down-regulated compared to hPSCs respectively (*Supplementary file 1*). Predictably, the most-downregulated genes were associated with undifferentiated hPSCs (e.g. NANOG, GDF3, POU5F1), anterior neurectoderm (OTX2) and lateral/ventral mesoderm (KDR). The vast majority of the top-upregulated genes were well-established drivers of axis elongation (e.g. *TBRA, CDX1/2, EVX1, MSGN1, TBX6*) and WNT/FGF/NOTCH/RA signalling pathway components, known to be expressed at high levels in the late PS/TB regions in vivo (e.g. *WNT3A/5B, RSPO3, FGF4/8, FGF17, HES7*) (*Figure 1B*, *Figure 1—figure supplement 1C,D*, *Supplementary file 1*). A large fraction of upregulated genes were transcriptional regulators (*Figure 1—figure supplement 1D*, *Supplementary file 1*) and we found that members of HOX PGs 1-9 were strongly differentially expressed between the two groups (*Figure 1C*, *Figure 1—figure supplement 1E*, *Supplementary file 1*). The upregulation of posterior thoracic PG(5-9) HOX transcripts as well as the presence of many transcripts (23/32) marking 'late' E9.5 mouse embryonic NMPs such as CYP26A1,

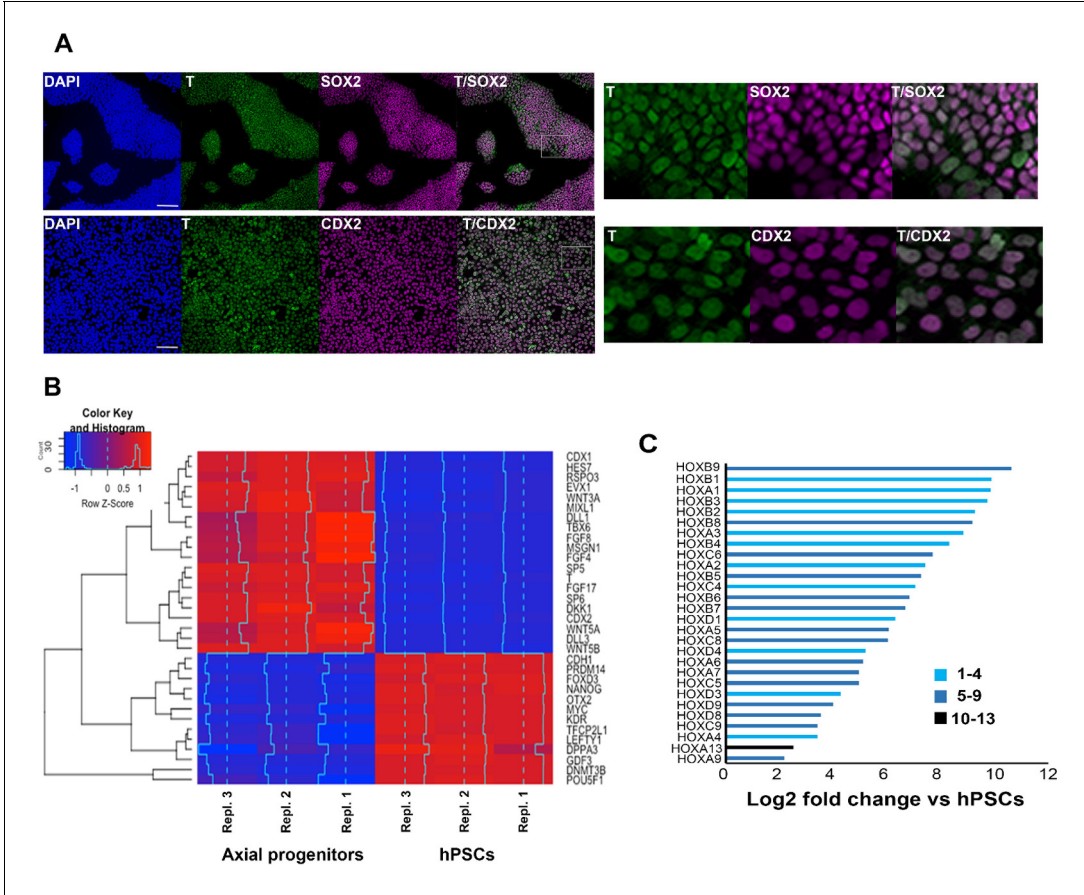

**Figure 1.** Transcriptome analysis of in vitro-derived human axial progenitors. (A) Immunofluorescence analysis of expression of indicated markers in day 3 hPSC-derived axial progenitors. Magnified regions corresponding to the insets are also shown. Scale bar = 100 µm. (B) Heatmap showing the expression values of selected markers in three independent axial progenitor and hPSC sample replicates. The expression values (FPKM) were scaled to the row mean. The color key relates the heat map colors to the standard score (z-score). (C) Induction of all significantly upregulated HOX transcripts in axial progenitors relative to hPSCs. Paralogous HOX groups corresponding to different axial levels such as cervical (groups 1–4), brachial/thoracic (groups 5-9)and lumbosacral (groups 10-13) are indicated.

DOI: https://doi.org/10.7554/eLife.35786.002

The following figure supplement is available for figure 1:

**Figure supplement 1.** RNASeq analysis of in vitro-derived axial progenitors.
DOI: https://doi.org/10.7554/eLife.35786.003

FGF17 and WNT5A (*Gouti et al., 2017*) suggest that day 3 WNT-FGF-treated hPSC cultures may correspond to a more developmentally advanced axial progenitor state. Overall, these data confirm our previous observations that treatment of hPSCs with WNT and FGF agonists gives rise to cultures resembling embryonic posterior axial progenitors.

## A neural crest signature in human axial progenitor cultures

We next sought evidence that might point to links between trunk NC and human axial progenitors. The RNAseq analysis revealed that a considerable number of genes known to mark the neural plate border and early NC in vivo ('NC/border' e.g. SOX9, PAX3, MSX1/2, SNAI1/2, ZIC1/3) were also significantly upregulated in axial progenitors (*Figure 2A*), a finding which was verified using quantitative real time PCR (qPCR) (*Figure 2—figure supplement 1A*). To exclude the possibility that the presence of such markers was the result of spontaneous differentiation of NM bipotent axial progenitors and their neural derivatives we examined their co-expression with T, a marker of both NMPs and prospective PXM. Immunostaining of d3 WNT-FGF-treated hPSC cultures showed that a considerable fraction (~40–60% in two different hPSC lines) of T+ cells also expressed the early NC

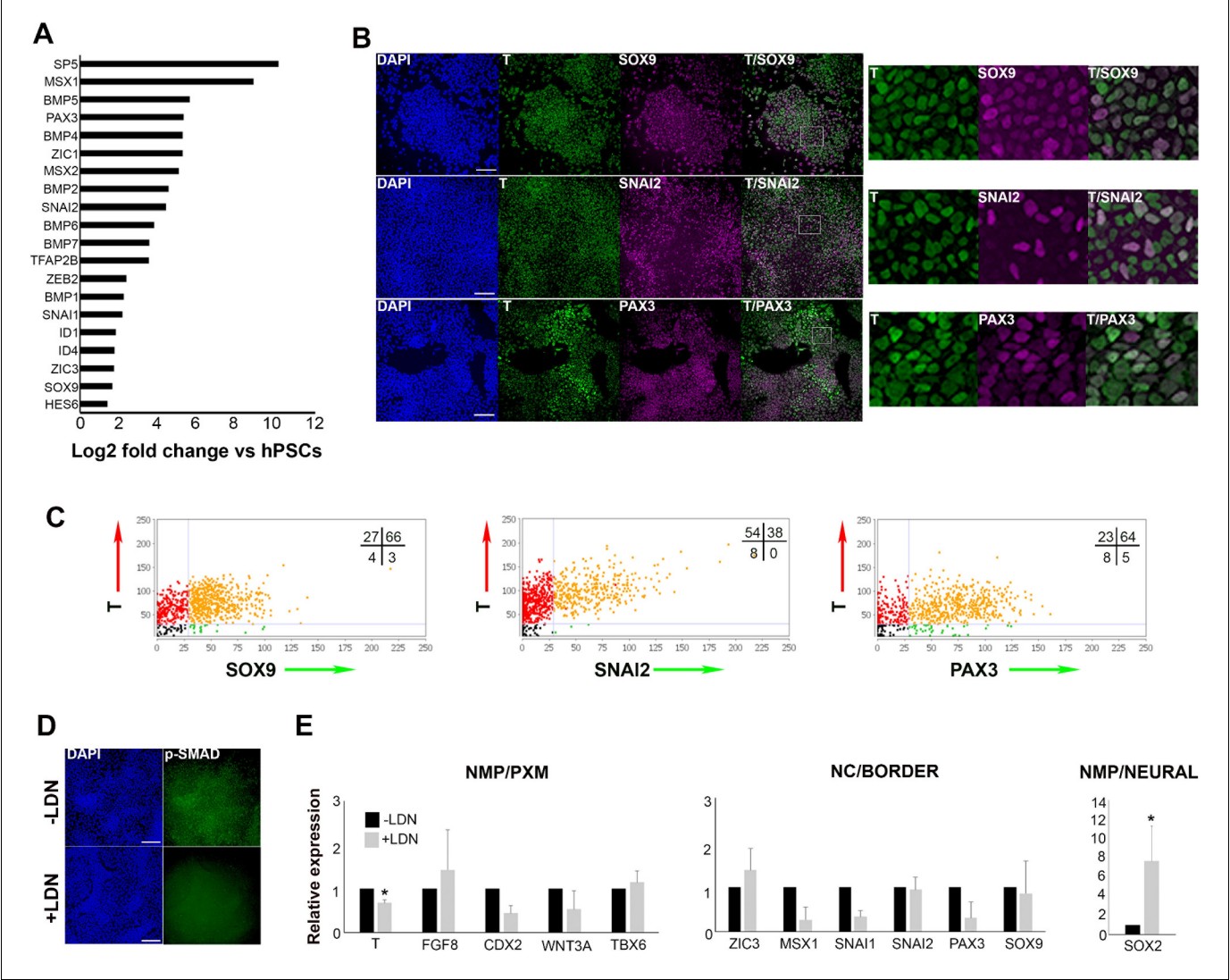

**Figure 2.** Human axial progenitor cultures exhibit a neural crest/border signature. (**A**) Log-fold induction of representative neural crest/neural plate border and BMP-associated transcripts in axial progenitors compared to hESCs. (**B**) Immunofluorescence analysis of expression of indicated markers in axial progenitors. Magnified regions corresponding to the insets are also shown. Scale bar = 100 μm. (**C**) Graphs showing the correlation of indicated NC/border markers with T in hPSC-derived (MasterShef7 line) axial progenitors. Each graph corresponds to one scored representative random field and reflects the results from the analysis of 3–5 random fields. Numbers in each quadrant represent percentages of cells belonging to different categories. Numbers of nuclei analysed in each graph: N = 785 (T-SOX9), N = 940 (T-SNAI2), N = 720 (T-PAX3). (**D**) Immunofluorescence analysis of expression of phosphorylated SMAD1/5 (p-SMAD) in the presence and absence of the BMP inhibitor LDN193189 (LDN). Scale bar = 100 μm. (**E**) qPCR expression analysis of indicated markers in axial progenitors in the presence (+) or absence (-) of LDN. Error bars = S.D. (n = 3). In all cases nuclei were counterstained with DAPI. PXM, paraxial mesoderm; NC, neural crest. *p<0.05, Paired t-test.

DOI: https://doi.org/10.7554/eLife.35786.004

The following source data and figure supplements are available for figure 2:

**Source data 1.** Raw data for *Figure 2*.
DOI: https://doi.org/10.7554/eLife.35786.007

**Figure supplement 1.** Dissecting the expression of neural crest/border markers in cultures of in vitro-derived axial progenitors.
DOI: https://doi.org/10.7554/eLife.35786.005

**Figure supplement 1—source data 1.** Raw data for *Figure 2—figure supplement 1*.
DOI: https://doi.org/10.7554/eLife.35786.006

markers/specifiers SOX9, SNAI2 and PAX3 (*Figure 2B and C*, *Figure 2—figure supplement 1B*). Moreover, SOX9-positive cells were found to express low or no MSGN1 or TBX6 (*Figure 2—figure supplement 1C*) suggesting that the upregulation of NC/border markers following WNT-FGF treatment of hPSCs was unlikely to reflect the presence of committed PXM cells given that some of these genes are also expressed in the mesoderm during axis elongation. Collectively, these findings indicate that a NC/border state arises within multipotent posterior axial progenitors which have not committed to a neural or mesodermal fate.

A number of both in vivo and in vitro studies have pointed to an optimal level of low/intermediate BMP signalling acting as an inducer of a NC/border character in conjunction with WNT and FGF (*Menendez et al., 2011*; *Hackland et al., 2017*; *Sasai et al., 2014*; *Tribulo et al., 2003*; *Streit and Stern, 1999*; *Patthey et al., 2009*; *Marchant et al., 1998*). We therefore examined whether hPSC-derived axial progenitor cultures exhibit endogenous BMP activity. RNAseq revealed that many BMP pathway -associated transcripts (BMP2/4/6/7 and ID1/4) were significantly upregulated compared to hPSCs (*Figure 2A*). Moreover, antibody staining showed expression of phosphorylated SMAD1/5, a readout of BMP activity (*Figure 2D*). This is extinguised upon treatment with the BMP antagonist LDN193189 (LDN) (*Cuny et al., 2008*) during the in vitro differentiation of hPSCs to day three axial progenitors (*Figure 2D*). Interestingly, BMP inhibition caused a decrease in the transcript levels of some NC/border-specific transcripts such as PAX3, SNAI1 and MSX1 (*Figure 2E*). We also observed a reduction in the expression of the NMP/late PS/TB markers CDX2, WNT3A and T (*Figure 2E*) although only the downregulation in the levels of the latter was statistically significant. This is consistent with a recent report showing that BMP signalling contributes to maintenance of T expression in the mouse tailbud (*Sharma et al., 2017*). We also found that the PXM specifier TBX6 remained relatively unaffected by LDN treatment suggesting that emergence of prospective PXM is not influenced by BMP inhibition (*Figure 2E*). By contrast, SOX2 expression was significantly increased upon LDN treatment (*Figure 2E*). Taken together these data indicate an association between endogenous BMP activity and the acquisition of a SOX2+ low border/NC identity by posterior axial progenitors while transition toward a SOX2+ high neural fate relies on BMP antagonism in vitro.

## In vitro-derived axial progenitors are a source of trunk neural crest cells

We reasoned that if posterior axial progenitors with NC/border features correspond to pioneer trunk NC precursors then they should be competent to generate definitive trunk neural crest when placed in an appropriate culture environment. We have recently reported a protocol for the efficient generation of anterior cranial NC cells from hPSCs involving the combined stimulation of WNT signalling, TGFβ signalling inhibition and moderate BMP activity via the parallel addition of BMP4 and the BMP type one receptor inhibitor DMH1 (*Hackland et al., 2017*).Culture of day 3 WNT-FGF-treated hPSCs under these NC-inducing conditions for 5–6 days gave rise to a high number (average percentage = 50% of total cells) of cells co-expressing the definitive NC marker SOX10 together with HOXC9, a readout of trunk axial identity (*Figure 3A–C*). We observed no nuclear staining above background intensity levels with the monoclonal HOXC9 antibody we employed in negative control undifferentiated hPSCs or ETS1+ NC cells generated using our cranial NC induction protocol (*Figure 3—figure supplement 1A,B*) (*Simoes-Costa and Bronner, 2016*). A large proportion of the cultures were also SOX9+ HOXC9+ further confirming a trunk NC character, whereas the percentage of neural cells marked by SOX1 expression remained very low throughout the course of the differentiation (*Figure 3—figure supplement 1C,D*). This may indicate that posterior NC progenitors do not progress through neural commitment but rather diverge from an earlier pre-neural, border-like stage reflecting previous reports which show that NC specification takes place prior to definitive neurulation (*Sasai et al., 2014*; *Leung et al., 2016*; *Basch et al., 2006*). Furthermore, during the transition toward trunk neural crest, the NMP/pre-neural marker NKX1-2 was rapidly extinguished followed shortly after by T, while CDX1 transcript levels declined more slowly (*Figure 3—figure supplement 1E*). By contrast, the expression of CDX2 and SOX9 was maintained at high levels throughout the course of differentiation of axial progenitors to trunk NC while SOX10 expression appeared only after day 7 of differentiation (Day 0 defined as the start of axial progenitor induction from hPSCs) (*Figure 3—figure supplement 1D–F*, data not shown).

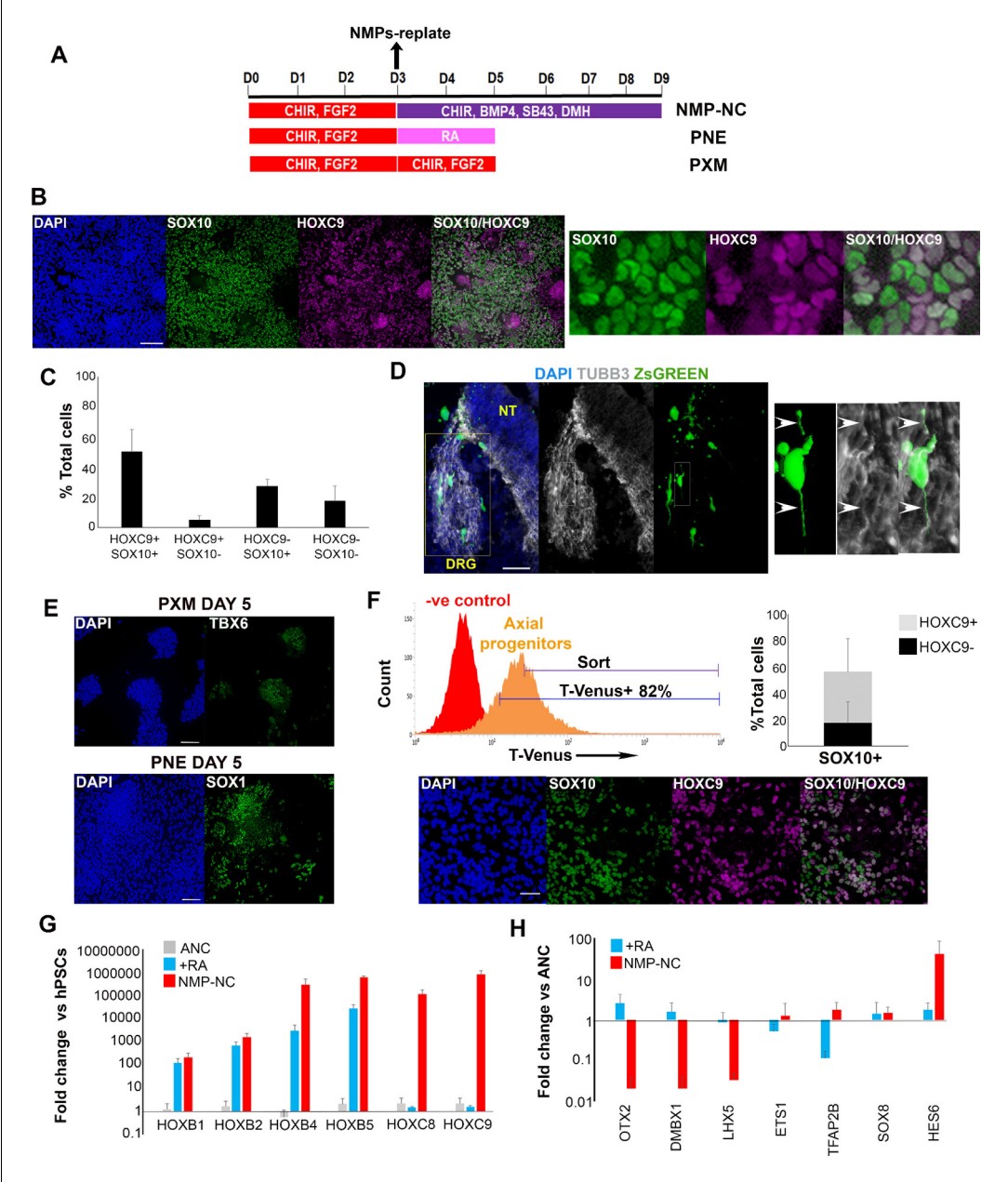

**Figure 3.** In vitro-derived axial progenitors generate trunk neural crest efficiently. (**A**) Diagram depicting the culture conditions employed to direct trunk NC, posterior neurectoderm (PNE) and paraxial mesoderm (PXM) differentiation from hPSC-derived axial progenitors. (**B**) Immunofluorescence analysis of the expression of the definitive NC marker SOX10 and the thoracic/trunk marker HOXC9 in trunk NC cells derived from axial progenitors after 8 days of differentiation (NMP-NC, see *Figure 3A*). A magnified region corresponding to the inset is also shown. Scale bar = 100 μm. (**C**) Quantification of cells marked by different combinations of HOXC9 and SOX10 expression in day eight trunk NC cultures derived from axial progenitors following image analysis. The data in the graph were obtained after scoring three random fields per experiment (two independent replicates) that is a total of 6 fields for two experiments and the error bars/standard deviation represent the variation across all 6 fields and two experiments. Total number of cells scored = 5366, average number of cells/field = 894, error bars = s.d. (**D**) Immunofluorescence analysis of ZsGREEN and TUBB3 expression in a section of a chick embryo grafted with ZsGREEN+ human axial progenitor-derived trunk NC cells. The DRG region is marked by a yellow box. The images on the right are magnifications of the region marked by the white inset within the DRG region. Arrowheads mark co-localisation of the ZsGREEN and TUBB3 proteins in a donor cell derived, DRG-localised neurite. V, ventral neural tube. Scale bar = 100 μm. (**E**) Immunofluorescence analysis of TBX6 (left) or SOX1 (right) expression in axial progenitors treated with CHIR-FGF2 (pro-PXM conditions) and RA (pro-PNE conditions) respectively. Scale bar = 100 μm. (**F**) Top left: Representative FACS histogram indicating the gated T-VENUS +hPSC derived axial progenitors as well as its flow-sorted fraction ('sort') which was subsequently plated in NC-inducing conditions. Top right: Average percentage of SOX10+ cells (in relation to HOXC9 expression) following 5 day differentiation of sorted T-VENUS+ axial progenitors in NC-inducing conditions, immunostaining and image analysis. The

*Figure 3 continued on next page*

*Figure 3 continued*

data in the graph were obtained after scoring 8–10 random fields per experiment (N = 5). The error bars/standard deviation represent the variation across all fields and five experiments. Error bars = s.d. Bottom: A representative field depicting immunofluorescence analysis of SOX10 and HOXC9 expression in NC cells derived from sorted T-VENUS+ axial progenitors. Scale bar = 100 μm. (G) qPCR expression analysis of indicated HOX genes in hPSC-derived anterior cranial (ANC), retinoic acid (RA)-treated NC (+RA), and axial progenitor-derived NC cells (NMP-NC) relative to hPSCs. Error bars = S.E.M. (n = 3). (H) qPCR expression analysis of indicated NC markers in +RA and axial progenitor-derived NC cells relative to untreated anterior cranial NC cells. Error bars = S.E.M. (n = 3).

DOI: https://doi.org/10.7554/eLife.35786.008

The following source data and figure supplements are available for figure 3:

**Source data 1.** Raw data for *Figure 3*.
DOI: https://doi.org/10.7554/eLife.35786.015
**Figure supplement 1.** Dynamics of trunk neural crest differentiation from axial progenitors.
DOI: https://doi.org/10.7554/eLife.35786.009
**Figure supplement 1—source data 1.** Raw data for *Figure 3—figure supplement 1*.
DOI: https://doi.org/10.7554/eLife.35786.010
**Figure supplement 2.** Characterisation of hPSC- derived axial progenitor differentiation products.
DOI: https://doi.org/10.7554/eLife.35786.011
**Figure supplement 2—source data 1.** Raw data for *Figure 3—figure supplement 2*.
DOI: https://doi.org/10.7554/eLife.35786.012
**Figure supplement 3.** Quantification of pluripotency marker expression in hPSC-derived axial progenitors.
DOI: https://doi.org/10.7554/eLife.35786.013
**Figure supplement 3—source data 1.** Raw data for *Figure 3—figure supplement 3*.
DOI: https://doi.org/10.7554/eLife.35786.014

## Characterisation of axial progenitor-derived trunk NC cells

We next tested the developmental potential of human axial progenitor-derived trunk NC cells. To this end, we grafted trunk NC cells derived from a human induced PSC (iPSC) line carrying a constitutive ZsGreen fluorescent reporter (*Lopez-Yrigoyen et al., 2018*) in or on top of the dorsal neural tube of Hamburger and Hamilton (HH) stage 10–11 chick embryos. We found that, following incubation for 2–3 days, grafted donor cells migrated out of the graft site (6/6 grafted embryos) (*Figure 3— figure supplement 2A*). Furthermore, the donor cells that had migrated the furthest consistently entered the dorsal root ganglia (DRG) and exhibited expression of DRG markers such as TUBB3 (*Shao et al., 2017*) (*Figure 3D*), ISL1 (*Ericson et al., 1992*) and SOX10 (*Ota et al., 2004*) (3/6 grafted embryos) (*Figure 3—figure supplement 2B,C*). These results suggest that human trunk NC generated from axial progenitors exhibits similar migratory behaviour/in vivo differentiation potential to its embryonic counterparts.

Since elevated BMP signalling appears to coincide with the acquisition of an early NC/border character by human axial progenitors (*Figure 2A and E*) we also examined whether inhibition of this pathway affects their ability to generate trunk NC. We found that LDN treatment of axial progenitors during their induction from hPSCs (i.e. between days 0–3 of differentiation) has no effect on subsequent trunk NC production (*Figure 3—figure supplement 2D*) indicating that early BMP activity alone is not the critical determinant of NC potency in this population. We also confirmed the NM potency of the starting axial progenitor cultures as treatment with high levels of FGF2-CHIR and RA led to the production of TBX6+/MSGN1 + PXM and SOX1+ spinal cord, posterior neurectoderm (PNE) cells respectively (*Figure 3A and E*, *Figure 3—figure supplement 2E,F*). Taken together these data suggest that hPSC-derived NM-potent axial progenitor cultures are competent to produce trunk NC at high efficiency.

To further confirm the lineage relationship between trunk NC cells and T+ axial progenitors we utilised a T fluorescent reporter hPSC line (*Mendjan et al., 2014*) and isolated, via flow cytometry, axial progenitors/NMPs expressing T-VENUS following 3 day treatment of hPSCs with FGF2 and CHIR for 3 days (*Figure 3F*, *Figure 3—figure supplement 2G*) in order to test their NC potential. T-VENUS+ axial progenitors exhibited no or very low (5% of total cells) expression of the definitive pluripotency markers OTX2 and NANOG (*Acampora et al., 2013*; *Osorno et al., 2012*) respectively (*Figure 3—figure supplement 3*) and hence are unlikely to be pluripotent. The small NANOG + T-VENUS+[low] fraction we detected (*Figure 3—figure supplement 3A,C*) probably reflects the

reported presence of *Nanog* transcripts in the gastrulation-stage posterior epiblast of mouse embryos (*Teo et al., 2011*). However, to avoid contamination from potentially pluripotent NANOG + T-VENUS+[low] cells, we sorted and analysed exclusively T-VENUS+[high] cells (*Figure 3F*). These were then plated in NC-inducing conditions for 5 days as described above (*Figure 3A*) and the acquisition of a trunk NC identity was examined. We found that almost 60% of the cells were SOX10+ and about a third of them also co-expressed HOXC9 (*Figure 3F*). This finding demonstrates that T + hPSC derived axial progenitors have the ability to generate efficiently SOX10+ neural crest and suggests that at least half of the trunk NC cells derived from bulk axial progenitor cultures (*Figure 3B,C*) originate from T-expressing cells.

Similar to established in vitro neural induction strategies, most current NC differentiation protocols aiming to generate posterior (e.g. trunk) cell populations from hPSCs rely on the caudalisation of an anterior ectodermal precursor via treatment with RA and/or WNT agonists (*Chambers et al., 2012*; *Huang et al., 2016*; *Oh et al., 2016*; *Fattahi et al., 2016*; *Denham et al., 2015*). Therefore, we compared our axial progenitor–based approach for generating trunk NC to a conventional strategy involving the generation of anterior cranial NC (ANC) precursor cells (*Hackland et al., 2017*) followed by RA addition in the presence of WNT and BMP signalling (*Figure 4A*). The axial identity of the resulting cells was assessed by qPCR assay of *HOX* transcripts corresponding to different levels along the A-P axis. In line with previous findings (*Huang et al., 2016*; *Fattahi et al., 2016*) RA-treated cells expressed high levels of *HOX* PG(1-5) members compared to untreated NC suggesting a posterior cranial and vagal/cardiac NC character (*Figure 3G*). However, efficient induction of trunk *HOXC8* and *9* transcripts was only achieved when posterior axial progenitors were employed as the starting population for NC generation (*Figure 3G*). Furthermore, axial progenitor-derived NC cells were marked by increased expression of the trunk NC marker *HES6*, but did not express the cranial markers *OTX2*, *DMBX1* and *LHX5* although they were positive for the 'late' cranial NC transcripts (*TFA2B*, *ETS1*, *SOX8*) (*Simoes-Costa and Bronner, 2016*) (*Figure 3H*). We thus conclude that posterior axial progenitors are the ideal starting population for efficiently generating trunk NC in vitro whereas RA treatment of anterior NC precursors predominantly produces posterior cranial and cardiac/vagal NC cells. These data also serve as evidence supporting the notion that trunk NC precursors are likely to arise within cells with axial progenitor/NMP features rather than a caudalised anterior progenitor. This is further supported by our T-VENUS sorting experiments showing that T-VENUS+[high]OTX2 negative axial progenitors are a source of trunk NC (*Figure 3F*, *Figure 3—figure supplement 3B,D*) and therefore the generation of these cells is unlikely to occur via 'caudalisation' of an anterior OTX2+ NC precursor.

## Efficient A-P patterning of human neural crest cells reveals molecular signatures of distinct axial identities

To further discern the identity of posterior NC subtypes induced either via RA treatment or an axial progenitor intermediate as well as identify unique associated molecular signatures we carried out analysis of the transcriptomes of NC cells arising under these conditions as well as those of their precursors using microarrays (*Figure 4A*). We found that axial progenitor-derived NC cells (NMP-NC d9) and their precursors (NMP-NC d6) grouped together and were distinct from a cluster containing d6 anterior cranial NC (ANC) and +RA NC cells and their common d3 progenitor (ANC d3) (*Figure 4B*, *Figure 4—figure supplement 1A*). Although the three final populations exhibited distinct transcriptional profiles (*Figure 4C*) they all expressed pan-NC genes including 'early' NC/border (MSX1/2, PAX3/7)- and 'late' NC (SOX10, SNAI1/2)-associated transcription factors (*Figure 4—figure supplement 1B,C*, *Supplementary file 2*). In line with our previous observations (*Figure 3G*), ANC cells failed to express any HOX transcripts, RA treatment induced anterior HOX genes and only WNT-FGF-treated hPSCs gave rise to NC cells positive for thoracic HOX PG(5-9) members (*Figure 4D*) reflecting an anterior cranial, posterior cranial/cardiac/vagal and trunk NC fate respectively. The simultaneous presence of anterior HOX PG(1-5) transcripts together with their more posterior group 6–9 counterparts in our trunk NC cultures would be expected in trunk NC due to their co-expression in the posterior neural tube/neural crest in E9.5 mouse embryos (*Gouti et al., 2017*; *Arenkiel et al., 2003*; *Bel et al., 1998*; *Glaser et al., 2006*). It might also result from the co-emergence of a separate population of cardiac/vagal NC cells during trunk NC differentiation due to the action of endogenous RA signalling since our microarray data revealed the upregulation of RA

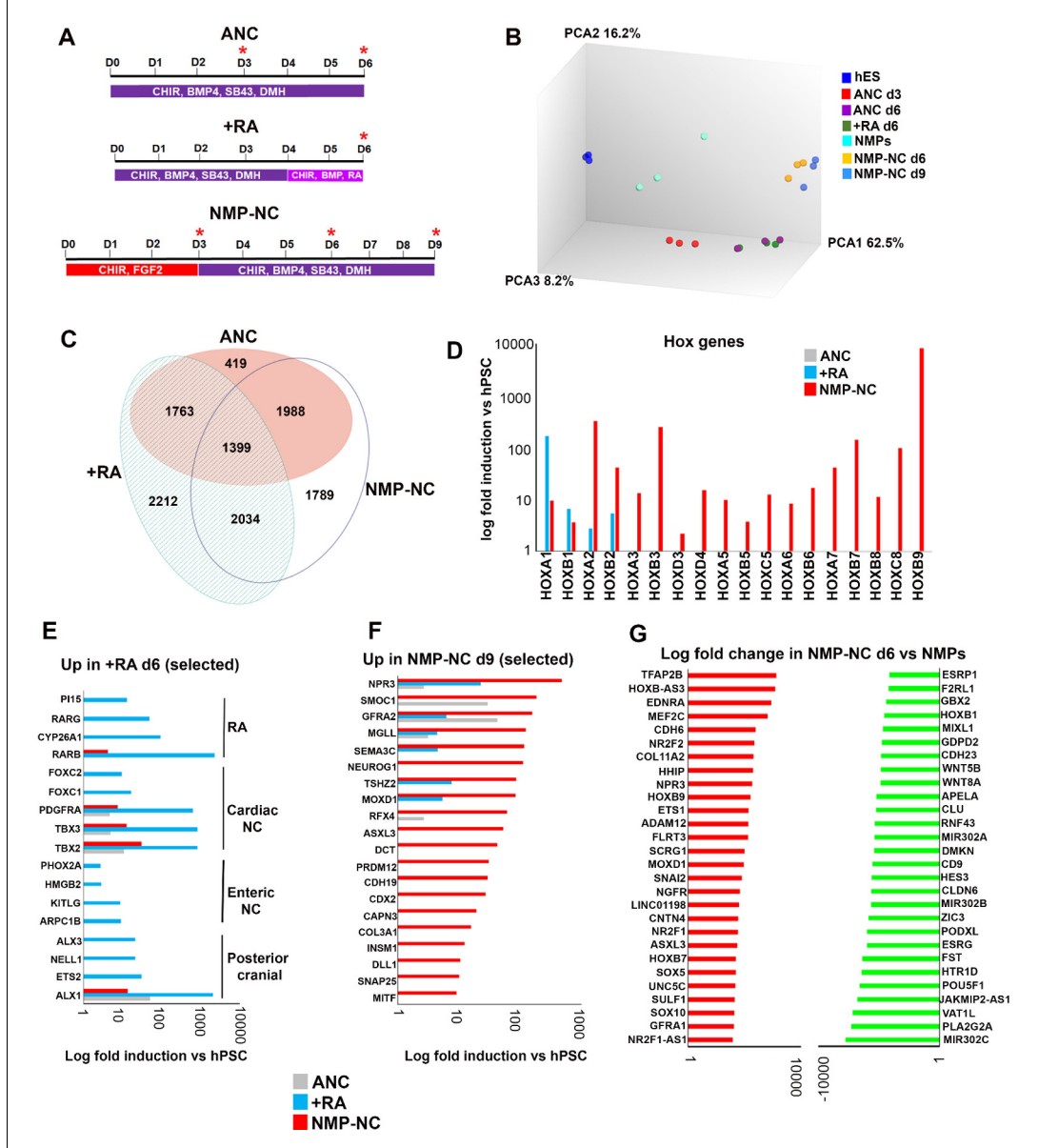

**Figure 4.** Transcriptome analysis of in vitro derived neural crest cells corresponding to distinct axial levels. (**A**) Diagrams showing the culture conditions employed for generating NC cells of distinct axial identities using hPSCs. Asterisks indicate the timepoints used for sample harvesting and transcriptome analysis. D, day of differentiation. ANC, Anterior neural crest. (**B**) Principal component analysis depicting variance between different samples used for microarray analysis (timepoints shown in A). (**C**) Venn diagram showing the overlap between all significantly upregulated ($\geq$2 fold relative to undifferentiated hESCs, FDR $\leq$ 0.05) in each indicated NC group. (**D**) Log fold induction of selected HOX genes in indicated NC populations relative to hPSCs. (**E**) Log fold induction of representative significantly upregulated ($\geq$2 fold relative to undifferentiated hPSCs, FDR $\leq$ 0.05) transcripts marking day 6 RA-treated NC cells. (**F**) Log fold induction of representative significantly upregulated ($\geq$2 fold relative to undifferentiated hPSCs, FDR $\leq$ 0.05) transcripts marking day nine axial progenitor-derived NC cells. (**G**) Log fold changes in the expression of the most-upregulated and most-downregulated transcripts in day six axial progenitor-derived NC precursors compared to d3 hPSC-derived axial progenitors.

DOI: https://doi.org/10.7554/eLife.35786.016

The following figure supplement is available for figure 4:

**Figure supplement 1.** Microarray analysis of hPSC-derived neural crest cells of distinct axial identities.

DOI: https://doi.org/10.7554/eLife.35786.017

signalling components in axial progenitor-derived trunk NC (*Supplementary file 2*) even though no exogenous RA was added to the differentiation medium.

The axial identity of the resulting posterior NC subtypes was further confirmed by the observation that some of the most-upregulated transcripts in +RA cells were established posterior cranial (e.g. ALX1/3 [*Lumb et al., 2017*]), cardiac (e.g. *FOXC1* and 2 (*Seo and Kume, 2006*); *PDGFRa* (*Tallquist and Soriano, 2003*); *TBX2/3* [*Mesbah et al., 2012*]) and vagal/enteric NC markers (*PHOX2A* (*Young et al., 1999*); *KITLG* (*Torihashi et al., 1996*); *ARPC1B* (*Iwashita et al., 2003*) (*Figure 4E*). In contrast, known trunk NC, sympathoadrenal and sympathetic/sensory neuron regulators such as CDX2 (*Sanchez-Ferras et al., 2016*), INSM1 (*Wildner et al., 2008*), NEUROG1 (*Perez et al., 1999*) (*Figure 4F*) and ISL1 (*Huber et al., 2013*) (*Supplementary file 2*) were induced only in NC cells derived from axial progenitors. We also identified *ASLX3*, the human homologue of the Drosophila polycomb protein *asx* (*Katoh and Katoh, 2004*) which has been recently linked to the developmental syndrome Bainbridge-Ropers (*Bainbridge et al., 2013*) as a novel trunk NC marker (*Figure 4F*, *Supplementary file 2*). Transcription factors specifically induced in anterior cranial NC cells included the forkhead gene *FOXS1* which has been shown to be expressed in mouse NC derivatives (*Heglind et al., 2005*) and *TCF7L2*, a WNT signalling effector which has been reported to harbour a NC-associated enhancer (*Rada-Iglesias et al., 2012*) (*Supplementary file 2*, *3*). Collectively these data support the idea that a mixed posterior cranial/vagal/cardiac NC character arises upon treatment of anterior NC precursors with RA whereas a *bona fide* trunk NC identity can be achieved only via an axial progenitor intermediate.

One of the most over-represented gene categories in all three axial NC subtypes were transcription factors and a common NC-specific transcription factor module was found to be expressed regardless of axial character (*Figure 4—figure supplement 1B*, *Supplementary file 3*, *4*). This included well-established NC/border regulators such as *PAX3/7*, *MSX2*, *SOX9/10*, *TFAP2A-C* and *SNAI1/2* (*Figure 4—figure supplement 1C*, *Supplementary file 3*, *4*). However, the expression levels of many of these transcription factors varied between the three groups (*Figure 4—figure supplement 1C*). The highest levels of HES6 and MSX1 were found in axial progenitor-derived trunk NC cells and their precursors whereas high PAX7 and SNAI1/SOX9 expression was more prevalent in the anterior cranial and RA-treated samples respectively (*Figure 4—figure supplement 1C*). Comparison of the day six trunk and d3 ANC precursor transcriptomes also revealed that expression of LHX5 and DMBX1 marks an anterior NC state whereas HES6 is associated exclusively with a trunk fate (*Figure 4—figure supplement 1C*) indicating that diversification of axial identity in NC cells starts at an early time point via the action of distinct molecular players.

## Distinct routes to posterior neural crest fates

To identify candidate genes mediating the gradual lineage restriction of trunk NC precursors present in axial progenitor cultures we compared the transcriptomes of d6 trunk NC precursors and day 3 WNT-FGF-treated hPSCs (='NMPs'). We found that dramatic global gene expression changes take place during the axial progenitor-trunk NC transition (*Figure 4G*, *Figure 4—figure supplement 1D*). Some of the most upregulated transcripts were the NC-specific TFAP2A/B, ETS1, SOX5 and SOX10 together with the established trunk NC specifier CDX2, the novel trunk NC marker ASLX3, the nuclear receptors NR2F1/2 and thoracic HOX genes (HOXB7, B9) (*Figure 4G*, *Supplementary file 4*). In contrast, signature axial progenitor transcription factors (*MIXL1*, *T*, *NKX1-2*) (*Figure 4G*, *Supplementary file 4*), anterior HOX genes (*HOXA1/B1*) and some WNT signalling components (*WNT8A/5B*) were significantly downregulated (*Figure 4G*). Thus, differentiation of trunk NC precursors appears to involve the transition from an axial progenitor-associated gene regulatory network to a NC-specifying one that incorporates factors which potentially act as general determinants of posterior cell fate (*CDX2, HOXB9*).

We also examined transcriptome changes during the transition from an anterior NC precursor state (ANC d3) to RA-posteriorised vagal/cardiac NC cells (+RA d6). The most-highly induced transcripts in posterior cranial/cardiac/vagal NC cells included the RA receptors beta and gamma (*RARb/g*) which have been involved in hindbrain and neural crest patterning (*Dupé et al., 1999*) and the T-box transcription factor *TBX2*, a marker of cardiac NC and in vitro derived vagal/enteric NC progenitors (*Fattahi et al., 2016*) (*Supplementary file 4*). Other upregulated transcripts included the planar cell polarity (PCP) component *PRICKLE1*, a regulator of cardiac NC cell function (*Gibbs et al., 2016*) and the TGFβ signalling-associated gene *TGFBI* (*Supplementary file 4*).

Anterior NC d3 precursor specific –transcripts included the border markers *PAX7* and *ZIC3* as well as the early cranial NC transcription factors *OTX2* and *LHX5* ( *Simoes-Costa and Bronner, 2016*) (*Supplementary file 4*). These results indicate that, in contrast to trunk NC cells, posterior crania/ cardiac/vagal NC cells arise from an anterior neural plate border precursor through posteriorisation under the influence of RA and possibly the non-canonical WNT and TGFβ pathways.

## Efficient in vitro generation of sympathoadrenal cells from axial progenitors

We next sought to determine whether trunk NC cells derived from axial progenitors are the optimal source of sympathoadrenal (SA) progenitors and their derivatives. The BMP and sonic hedgehog (SHH) signalling pathways have been shown to be critical for the specification of these lineages from NC cells (*Oh et al., 2016*; *Schneider et al., 1999*; *Morikawa et al., 2009*). Therefore we cultured d8 trunk NC cells generated from axial progenitors carrying a GFP reporter within the SA/sympathetic neuron regulator PHOX2B locus (*Oh et al., 2016*; *Pattyn et al., 2000*), in the presence of BMP and sonic hedgehog (SHH) signalling agonists (*Figure 5A*). GFP expression was assayed after 4 days of culture of trunk NC cells in BMP4 and SHH agonists (i.e. day 12 of differentiation) (*Figure 5B*). FACS analysis revealed that the majority of cells were PHOX2B expressing (average percentage from four independent experiments = 73.5%, s.d. = 6.3) (*Figure 5B*) and a large proportion of them were also positive for the early SA progenitor marker ASCL1 (*Hirsch et al., 1998*) indicating that they had acquired a symphathoadrenal identity (*Figure 5—figure supplement 1A*). Further maturation of the resulting SA progenitors in the presence of neurotrophic factors (BDNF, GDNF and NGF) resulted in the induction of a high yield of sympathetic neurons/progenitors co-expressing PHOX2B together with the sympathetic neuron regulator GATA3 (*Tsarovina et al., 2010*) (average of 40% of total cells), ASCL1 (63%) and the SA differentiation regulator *ISL1* (64%) (*Figure 5C and D*). At later stages of differentiation almost all PHOX2B-GFP + cells co-expressed ASCL1 further confirming a gradual transition from an early ASLC1+PHOX2B- progenitor to a more 'mature' double positive state (data not shown). A high proportion of the resulting cells also expressed the catecholamine production-associated enzyme/sympathetic neuron markers tyrosine hydroxylase (TH) (*Figure 5D*) together with dopamine- β-hydroxylase (DBH) (*Ernsberger et al., 2000*) (*Figure 5—figure supplement 1B*). Furthermore, the cultures widely expressed the peripheral nervous system marker PERIPHERIN (PRPH) (*Troy et al., 1990*) together with the trunk axial marker HOXC9 (*Figure 5—figure supplement 1C*). We also detected dramatic induction of *GATA3*, *ASCL1*, *TH* and *PHOX2B* transcripts (between 1000 and 1,000,000-fold) as well as other SA lineage markers such as *GATA2, DBH* and to a lesser extent *PHOX2A* using qPCR (*Figure 5E*).

We further examined the physiological properties of the sympathetic neurons produced from human axial progenitor derived-trunk NC using patch clamp recording. Following depolarising current injection, the neurons were found to fire either a single action potential (AP) at the stimulus onset (type I) or a sequence of 'regenerative' APs (type II) (*Figure 5F*). Similar electrophysiological responses have been previously reported to be indicative of in vitro derived sympathetic neurons (*Oh et al., 2016*). Furthermore, we found that the outward potassium currents in the Type I cells activated at significantly more hyperpolarised potentials than those in the Type II cells, which would be the likely cause of the different spiking characteristics observed in these cells (*Figure 5—figure supplement 1D*). We also confirmed that our sympathetic neurons secrete the catecholamines dopamine (DA) and norepinephrine (NE) further confirming their functionality (*Figure 5G*). Together, these results suggest that the most efficient route toward the production of sympathoadrenal cells and functional sympathetic neurons from hPSCs relies on the induction of posterior axial progenitors.

## Discussion

Despite progress in the optimisation of current NC differentiation protocols the in vitro generation of trunk NC cells from hPSCs remains challenging and requires FACS-sorting of selected progenitor subpopulations, a time-consuming and laborious process associated with increased cell death. This bottleneck prevents the dissection of the mechanisms directing human NC emergence at different axial levels as well as the efficient isolation of cell types for modelling trunk NC-specific neurocristo- pathies such as neuroblastoma. Previous work in amniote embryos suggested that posterior (trunk/

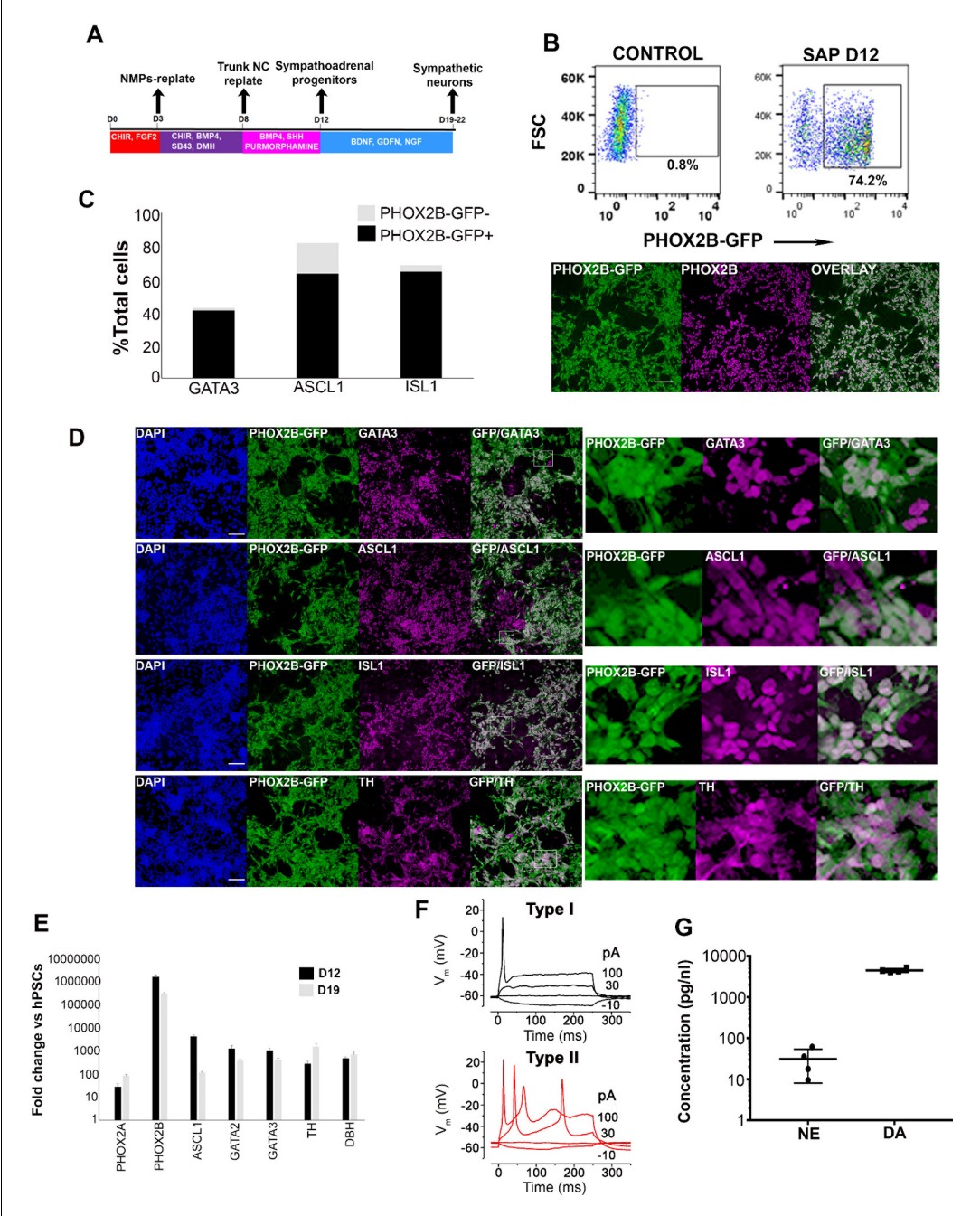

**Figure 5.** Axial progenitor-derived trunk neural crest is an optimal source of sympathoadrenal cells. (**A**) Diagram depicting the culture conditions employed to direct axial progenitors ('NMPs') toward trunk NC and subsequently sympathoadrenal progenitors (SAP) and sympathetic neurons. (**B**) FACS analysis of PHOX2B-GFP expression in SAP cells derived from axial progenitors as shown in A. Below: Immunofluorescence analysis of PHOX2B-GFP and PHOX2B protein expression following antibody staining. Scale bar = 100 µm. (**C**) Quantification of d18 differentiated cells positive for the indicated markers in relation to PHOX2B-GFP expression following antibody staining. In each case four randomly selected representative fields were used to obtain the average number of cells/marker. Total numbers of cells scored: GATA3 (N = 3003), ASCL1 (N = 2575), ISL1 (N = 2963). (**D**) Immunofluorescence analysis of PHOX2B-GFP together with the indicated markers in day 18 differentiated SAP/sympathetic neurons derived from axial progenitors as shown in A. Scale bar = 100 µm. (**E**) qPCR expression analysis of indicated SAP/sympathetic neuron markers in d12 and d18 cultures. Error bars = S.E.M. (n = 3). (**F**) Voltage responses of hPSC-derived sympathetic neurons (after day 19 of differentiation) to current injection. Type I and Type II cells were current clamped and hyperpolarising (negative) and depolarising (positive) current steps were applied (the current injected is shown next to the traces). The resulting membrane potential responses of the cells to these current injections are shown, the traces are overlaid. (**G**) Analysis

*Figure 5 continued on next page*

*Figure 5 continued*

of catecholamine production in hPSC-derived sympathetic neurons (after day 19 of differentiation) using a commercial ELISA kit (n = 2). NE, norepinephrine; DA dopamine.

DOI: https://doi.org/10.7554/eLife.35786.018

The following source data and figure supplement are available for figure 5:

**Source data 1.** Raw data for *Figure 5*.

DOI: https://doi.org/10.7554/eLife.35786.020

**Figure supplement 1.** Characterisation of axial progenitor-derived sympathoadrenal progenitors and sympathetic neurons.

DOI: https://doi.org/10.7554/eLife.35786.019

lumbosacral) NC cells arise independently from their anterior counterparts, within a pool of axial progenitors localised near the primitive streak and the tailbud during axis elongation (*Catala et al., 1995*; *Schoenwolf et al., 1985*; *Schoenwolf and Nichols, 1984*; *Wymeersch et al., 2016*; *Tzouanacou et al., 2009*). Here we utilised these findings and exploited our ability to induce T+ NM potent axial progenitors from hPSCs in order to use them as the optimal starting point for the efficient in vitro derivation of trunk NC (~50% HOXC9+ SOX10+), SA progenitors (~70% PHOX2B-GFP +) and functional sympathetic neurons without the use of FACS sorting. This strategy represents a considerable improvement over current approaches, which typically yield 5–10% PHOX2B-GFP + cells (*Oh et al., 2016*) and is in line with a recent study reporting the successful production of chromaffin-like cells through the use of an NC-induction protocol which transiently produces T + SOX2+ cells (*Denham et al., 2015*; *Abu-Bonsrah et al., 2018*).

We show that, similar to neural cells a HOX-positive posterior identity is acquired by human NC cells via two distinct routes: posterior cranial/vagal/cardiac HOX PG(1-5)+ NC cells emerge through the RA/WNT-induced posteriorisation of a default anterior precursor, reflecting Nieuwkoop's 'activation-transformation' model, whereas HOX PG(5-9)+ trunk NC cells arise from a separate WNT/FGF-induced posterior axial progenitor exhibiting caudal lateral epiblast/NMP features mixed with a neural plate border/neural crest identity (*Figure 6*). This finding offers an explanation for the failure of current RA posteriorisation-based in vitro differentiation protocols (*Huang et al., 2016*; *Fattahi et al., 2016*) to yield high numbers of HOX9+ trunk NC cells and should serve as the conceptual basis for the design of experiments aiming to generate NC cells of a defined A-P character from hPSCs.

Our data indicate that a subpopulation of in vitro derived human axial progenitors acquires border/early NC characteristics in response to the WNT and FGF signals present in the differentiation culture media, and possibly under the influence of autocrine BMP signalling. This is in line with bulk and single cell transcriptome data showing that mouse embryonic axial progenitors/NMPs express border and early NC markers (*Gouti et al., 2017*; *Koch et al., 2017*). Furthermore, our data reflect findings in the chick embryo showing that an 'unstable', pre-neural plate border domain, potentially defined by the co-localisation of pre-neural (*Nkx1-2)* (*Delfino-Machín et al., 2005*) and border markers such as *Pax3* (*Bang et al., 1997*) and *Msx1*, arises in the avian embryonic caudal lateral epibllast in response to autocrine BMP, FGF (*Streit and Stern, 1999*) and possibly WNT signalling (*LaBonne and Bronner-Fraser, 1998*). We also found that CDX2 expression is maintained at high levels during the generation of trunk NC indicating that this transcription factor might be critical in inducing an NC character in axial progenitors. CDX2 has been shown, together with β-catenin, to bind and activate neural plate border/early NC specifiers such as MSX1 and ZIC1 (*Sanchez-Ferras et al., 2016*; *Funa et al., 2015*) and, intriguingly, ChiP-Seq data from in vitro-derived mouse NMPs have revealed that many NC/border genes are direct targets of CDX2 often jointly with T (e.g. PAX3, SOX9, ZIC3) (*Amin et al., 2016*). Collectively these findings raise the possibility that β-catenin and CDX2, in conjunction with FGF/BMP signalling, may be critical for the establishment of a NC/border identity in T+ axial progenitors and further work is required to test this hypothesis.

We provide evidence that BMP/WNT treatment of human axial progenitors promotes the induction of a definitive trunk NC state. This transition appears to coincide with the progressive extinction of key axial progenitor genes and their replacement by a battery of NC-specific transcription factors such as TFAP2B, SOX10, NR2F2 and NR2F1 while the levels of some 'common' axial progenitor-NC markers (e.g. SOX9, PAX3, TFAP2A and SNAI2) remain high (*Figure 4*, *Figure 3—figure supplement 1*, *Supplementary file 4*). TFAP2A has been previously reported to act as a master NC

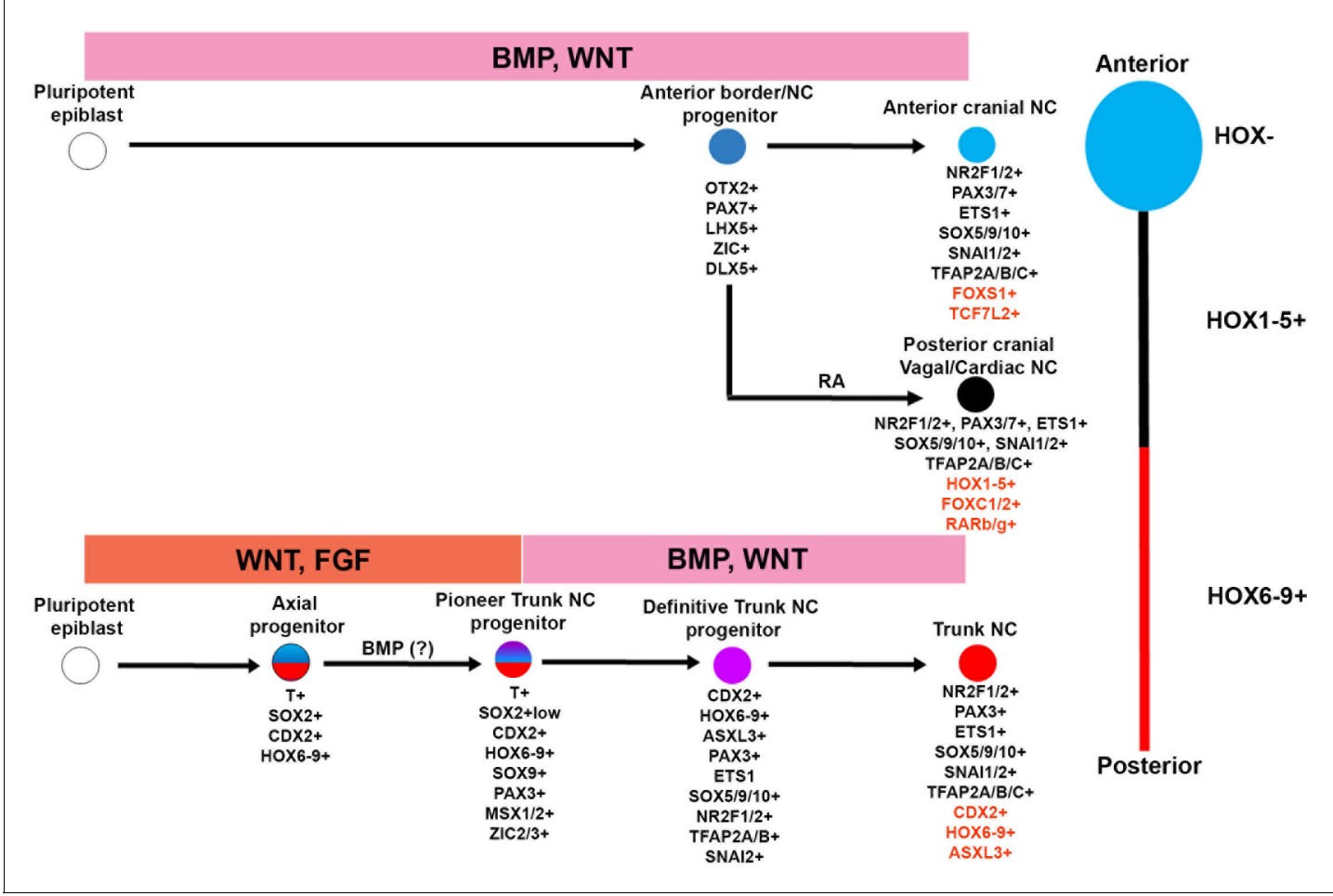

**Figure 6.** A-P patterning of in vitro derived human NC cells. Diagrammatic model summarising our findings on the in vitro generation of NC subtypes of distinct A-P identity from hPSCs. Examples of unique genes that were found to mark each NC population exclusively are shown in red.
DOI: https://doi.org/10.7554/eLife.35786.021

transcription factor whose binding on key enhancers, together with NR2F1/2, appears to initiate transcription of NC-specific genes (*Rada-Iglesias et al., 2012*). This finding raises the possibility that these transcription factors are the molecular drivers of the transition from an early posterior axial progenitor state to a lineage-restricted trunk NC fate in response to BMP/WNT. How is a trunk axial character specified? Our transcriptome analysis data suggest that, at least in vitro, a generic trunk identity is first installed during the emergence of multipotent CDX2+trunk HOX+axial progenitors under the influence of WNT-FGF activity and then 'converted' into posterior NC through the progressive accumulation of neural plate border/definitive NC markers within BMP-responsive cells. The posterior character of these incipient trunk NC precursors and their progeny is likely to be maintained via continuous CDX2 expression and further potentiation of trunk HOX activities as indicated by the expression profiles of these genes in both d6 precursor and d9 trunk NC cells (*Figure 4*, *Supplementary file 2–4*) and their reported roles in trunk NC specification. However, other trunk NC-specific regulators may also be involved in this process and loss-/gain of-function approaches are required to dissect their exact involvement in programming trunk identity.

# Materials and methods

**Key resources table**

*Continued on next page*

*Continued*

| Reagent type (species) or resource | Designation | Source or reference | Additional information |
| --- | --- | --- | --- |
| Reagent type (species) or resource | Designation | Source or reference | Additional information |
| cell line (Homo Sapiens) | T-VENUS | 68 | Parental hES cell line = H9 |
| cell line (Homo Sapiens) | SOX10-GFP | 15 | Parental hES cell line = H9 |
| cell line (Homo Sapiens) | PHOX2B-GFP | 18 | Parental hES cell line = H9 |
| cell line (Homo Sapiens) | MSGN1-VENUS | Unpublished | Not previously described, parental line = NCRM1 iPSCs (source = NIH) |
| cell line (Homo Sapiens) | Sox2-GFP | 44 | Parental hES cell line = Shef4 |
| cell line (Homo Sapiens) | MIFF1 | 102 | iPSC line from healthy donor |
| cell line (Homo Sapiens) | SFCi55-ZsGr | 64 | iPSC line from healthy donor containing a constitutive fluorescent ZsGreen reporte |
| cell line (Homo Sapiens) | MasterShef7 | 44 | Wild type hES cell line |

## Cell culture and differentiation

We employed the following hPSC lines: a Shef4-derived Sox2-GFP reporter hESC line (*Gouti et al., 2014*), the H9-derived T-VENUS (*Mendjan et al., 2014*), SOX10-GFP (*Chambers et al., 2012*) and PHOX2B-GFP (*Oh et al., 2016*) reporter hESC lines, the MSGN1-VENUS reporter hiPSC line, the wild type Mastershef7 hESC line (*Gouti et al., 2014*) and an iPSC line (MIFF-1) derived from a healthy individual (*Desmarais et al., 2016*). Chick embryo grafting experiments employed an iPSC line containing a ZsGreen reporter cassette (SFCi55-ZsGr iPSCs) (*Lopez-Yrigoyen et al., 2018*). The MSGN1-Venus reporter line was generated by Transposon mediated BAC transgenesis using protocols described by (*Rostovskaya et al., 2012*). In brief, a human BAC (RP11-12L16) with piggyBac transposon repeats flanking the bacterial backbone and with Venus inserted directly after the initiating methionine of MSGN1 was transfected together with a piggyBac Transposase into NCRM1 iPSCs. Use of hES cells has been approved by the Human Embryonic Stem Cell UK Steering Committee (SCSC15-23). All cell lines were tested mycoplasma negative. Cells were cultured in feeder-free conditions in either Essential 8 (Thermo Fisher) or mTeSR1 (Stem Cell Technologies) medium on laminin 521 (Biolamina) or vitronectin (Thermo Fisher). All differentiation experiments were carried out in at least three different hPSC line. For NMP/axial progenitor differentiation hPSCs were dissociated using PBS/EDTA and plated at a density of 55,000 cells/cm$^2$ (density optimised for 12-well plates) on fibronectin (Sigma) or vitronectin (Thermo Fisher)-coated wells, directly into NMP-inducing medium containing CHIR99021 (Tocris), FGF2 (20 ng/ml, R and D) and ROCK inhibitor Y-27632 (Tocris or Generon) for the first only day (10 µM, Tocris). We observed some variation in terms of induction of T + SOX2+NMPs both between hPSC lines and also batches of CHIR99021 and thus the concentration of the latter was varied between 3–4 µM. BMP inhibition was carried out using LDN193189 (Tocris) at 100 nM. For trunk NC differentiation day 3 hPSC-derived axial progenitors were dissociated using accutase and re-plated at a density 30,000 cells /cm$^2$ on Geltrex (Thermo Fisher)-coated plates directly into NC-inducing medium containing DMEM/F12 (Sigma), 1x N2 supplement (Thermo Fisher), 1x Non-essential amino acids (Thermo Fisher) and 1x Glutamax (Thermo Fisher), the TGFb/Nodal inhibitor SB431542 (2 µM, Tocris), CHIR99021 (1 µM, Tocris), DMH1 (1 µM, Tocris), BMP4 (15 ng/ml, Thermo Fisher) and Y-27632 (10 µM, Tocris/Generon). The medium was replaced at days 5 and 7 of differentiation but without the ROCK inhibitor and trunk NC cells were analysed either at day 8 or 9. For cranial neural crest differe ntiation hPSCs were dissociated using accutase and plated under the same NC-inducing conditions as described above for 5–6 days. For posterior cranial/

vagal/cardiac NC generation d4 differentiated anterior NC progenitors induced as described above were treated with retinoic acid (1 µM, Tocris) in the presence of the NC-inducing medium till day 6 of differentiation. For sympathoadrenal progenitor (SAP) differentiation d8 trunk NC cells were re-suspended at a density of 200–300,000 cells /cm$^2$ on Geltrex (Thermo Fisher)-coated plates directly into medium containing BrainPhys neuronal medium (Stem Cell Technologies), 1x B27 supplement (Thermo Fisher), 1x N2 supplement (Thermo Fisher), 1x Non-essential amino acids (Thermo Fisher) and 1x Glutamax (Thermo Fisher), BMP4 (50 ng/ml, Thermo Fisher), recombinant SHH (C24II) (50 ng/ml, R and D) and purmorphamine (1.25–1.5 µM, Millipore or Sigma) and cultured for 4 days (d12 of differentiation). For further sympathetic neuron differentiation d12 SAP cells were switched into a medium containing BrainPhys neuronal medium (Stem Cell Technologies), 1x B27 supplement (Thermo Fisher), 1x N2 supplement (Thermo Fisher), 1x Non-essential amino acids (Thermo Fisher) and 1x Glutamax (Thermo Fisher), ascorbic acid (200 µM, Cat. no: A8960, Sigma), NGF (10 ng/ml, Peprotech), BDNF (10 ng/ml, Peprotech) and GDNF (10 ng/ml, Peprotech). For paraxial mesoderm differentiation d3 axial progenitor cultures were treated with accutase and replated at a density of 45,000/cm$^2$ on 12-well Geltrex-coated plates in N2B27 containing FGF2 (40 ng/ml, R and D) and CHIR99021 (8 µM, Tocris) for two days. For neural differentiation d3 axial progenitor cultures were treated with accutase and replated at a density of 45,000/cm$^2$ on 12-well Geltrex-coated plates in N2B27 containing 100 nM retinoic acid (Tocris) for 2–3 days.

## RNA sequencing
### Sample preparation
For RNA sequencing we employed hESCs or axial progenitors (Shef4-derived Sox2-GFP reporter hESC line) following culture on fibronectin in FGF2 (20 ng/ml) and CHIR99021 (3 µM). Total RNA from NMPs and hESCs was harvested using the RNeasy kit (Qiagen) according to the manufacturer's instructions.

### Library preparation/sequencing
Total RNA was processed according to the TruSeq protocol (Illumina). Three separate RNA libraries (biological replicates) were barcoded and prepared for hPSCs and D3 axial progenitors. Library size, purity and concentration were determined using the Agilent Technologies 2100 Bioanalyzer. For sequencing, four samples were loaded per lane on an Illumina Genome Analyzer Hiseq2500.

### RNAseq quality control and mapping
The quality of raw reads in fastq format was analyzed by FastQC (http://www.bioinformatics.babraham.ac.uk/projects/fastqc). Adapter contamination and poor quality ends were removed using Trim Galore v. 0.4.0 (Babraham Bioinformatics - Trim Galore! Available at: http://www.bioinformatics.babraham.ac.uk/projects/trim_galore/). Single-end clean reads were aligned to the human reference genome (hg38 assembly) using Tophat2 v2.0.13 (*Kim et al., 2013*).

### RNA seq data analysis
Read alignments were sorted with SAMtools v1.1 before being counted to genomic features by HTSeq version 0.6.0 (*Anders et al., 2015*). The average overall read alignment rate across all samples was 94.3%. Differential gene expression was performed using DESeq2 version 1.16.1 (*Love et al., 2014*). in R version 3.3.3. Genes were considered significantly differentially expressed (DE) with a Benjamini-Hochberg adjusted pvalue <= 0.05 and a log2FoldChange> |1|. Gene Ontology (GO) biological processes (BP) enrichment analysis was carried out for DE genes using the DAVID gene ontology functional annotation tool (https://david.ncifcrf.gov/)

(*Huang et al., 2009a*; *Huang et al., 2009b*) with default parameters. We considered as significant terms having a FDR adjusted pvalue <= 0.05, which is derived from a modified Fisher's exact test.

## Microarrays
### Sample preparation and processing
Samples (derived from SOX10-GFP hES cells) were prepared according to the Affymetrix WT Plus protocol for Gene Chip Whole Transcript Expression Arrays. Briefly 200 ng of high quality total RNA, (RNA integrity number (RIN) greater than 9), was converted to double stranded cDNA with the

introduction of a T7 polymerase binding site. This allowed the synthesis of an antisense RNA molecule against which a sense DNA strand was prepared. The RNA strand was digested and the resulting single stranded DNA fragmented and biotin labelled. Along with appropriate controls the labelled fragmented DNA was hybridised to Affymetrix Clariom D arrays overnight using the Affymetrix 640 hybridisation oven; 16 hr with rotation at 60 rpm at 45°C. The arrays were washed and stained according to standard protocols which allowed the introduction of streptavidin-phycoerythrin in order to generate a fluorescent signal from the hybridised biotinylated fragments. The washed and stained arrays were scanned using the Affymetrix 3000 7G scanner with autoloader. The generated CEL files were taken forward for analysis.

## Data analysis

Data were analysed using the Affymetrix Transcriptome Analysis Console 4.0 software. Analysis of Expression (Gene + Exon) was used to generate lists of all differentially expressed genes showing >2; <-2 fold Log Change and p<0.05. For the distance matrix (*Figure 4—figure supplement 1*), Exploratory Grouping analysis was used. Log$_2$ normalised intensity data values were mapped in R using the package 'pheatmap' with correlation clustering by gene. Gene ontology analysis was carried out using the ToppGene suite (https://toppgene.cchmc.org/enrichment.jsp) (*Chen et al., 2009*). Area proportional 3-Venn diagrams were drawn using the eulerApe software (*Micallef and Rodgers, 2014*).

## Quantitative real time PCR

Total RNA from different samples was harvested using the RNeasy kit (Qiagen) according to the manufacturer's instructions and digested with DNase I (Qiagen) to remove genomic DNA. First strand cDNA synthesis was performed using the Superscript III system (Thermo Fisher) using random primers. Quantitative real time PCR was carried out using the Applied Biosystems QuantStudio 12K Flex thermocycler together with the Roche UPL system. Statistical significance was calculated using GraphPad Prism (GraphPad Software Inc, USA). Primer sequences are shown in *Supplementary file 5*.

## Flow cytometry

Flow cytometry was carried out using a BD FACSJAZZ cytometer (BD Biosciences). Cells were lifted into a single cell suspension using Accutase (as previously described) and resuspended in FACS buffer (DMEM with 10% v/v FCS) to neutralise Accutase before centrifugation at 1100 rpm/4 min. Cells were then resuspended in FACs buffer at 1 × 10⁶ cells/ml. A GFP baseline was set using unmodified wild type control cells. Cells that had been sorted were subsequently reanalysed for purity checking with a minimum accepted purity of 95%.

## Chick embryo grafting

Fertilised Bovan brown chicken eggs (Henry Stewart and Co., Norfolk, UK) were staged according to *Hamburger and Hamilton (1951)*. On day 8 of trunk neural crest differentiation (*Figure 3A*), cells were plated at concentrations of 1000–5000 cells/cm$^2$ as hanging drops in DMEM F12 media (Sigma) supplemented with N2 supplement (Thermo Fisher), MEM Non Essential Amino Acids (Thermo Fisher), Glutamax (Thermo Fisher) and Poly Vinyl Alcohol (4 mg/ml) (Sigma Aldrich). After aggregating overnight at 37°C, cell clumps (diameter =~ 50–100 µm) were implanted into or on top of the neural tube of HH stage 10–12 chick embryos. The roof plate was lesioned to accommodate transplanted tissue at the level of newly forming somites. Embryos were harvested 2–3 days post transplantation (HH stages 18–24).

## Immunofluorescence

Cells were fixed for 10 min at 4°C in 4% paraformaldehyde (PFA) in phosphate buffer saline (PBS), then washed in PBST (PBS with 0.1% Triton X-100) and treated with 0.5 M glycine/PBS to quench the PFA. Blocking was then carried for 1–3 hr in PBST supplemented with 3% donkey serum/1% BSA at room temperature or overnight at 4°C. Primary and secondary antibodies were diluted in PBST containing in PBST supplemented with 3% donkey serum/1% BSA. Cells were incubated with primary antibodies overnight at 4°C and with secondary antibodies at room temperature for 2 hr in the dark.

Cell nuclei were counterstained using Hoechst 33342 (1:1000, Thermo Fisher) and fluorescent images were taken using the InCell Analyser 2500 system (GE Healthcare). Chick embryos were fixed in 4% PFA for 2–3 hr at 4°C and left in 30% sucrose solution overnight at 4°C. Chick embryos were mounted in OCT (VWR 361603E) and transverse sections (15–20 μm) were taken using a cryostat. Immunostaining of sections was performed as previously described (*Placzek et al., 1993*). Briefly, overnight incubation with the primary antibody at 4°C was followed by short washes in PBS/0.1% Triton X-100 solution (PBST), one hour incubation with the secondary antibody and further PBST washes. Slides were mounted in Fluoroshield with DAPI (Sigma) and imaged on the InCell Analyser 2200 (GE Healthcare). We used the following antibodies: anti-T (1:200; AF2085, R and D, RRID: AB_2200235 or 1:200; ab209665, Abcam), anti-SOX2 (1:200; ab92494, Abcam, RRID:AB_10585428), anti-SOX9 (1:200; 82630, CST, RRID:AB_2665492), anti-SNAI2 (1:400; C19G7, CST, RRID:AB_2239535), anti-PAX3 (1:50; DSHB), anti-phosphoSMAD1/5/9 (1:100; D5B10, CST, RRID:AB_2493181), anti-SOX10 (1:200; D5V9L, CST), anti-SOX1 (1:100; AF3369, R and D, RRID:AB_2239879), anti-TBX6 (1:50, AF4744, RRID:AB_2200834) anti-TH (1:1000; T1299, SIGMA, RRID:AB_477560), HOXC9 (1:50; ab50839, Abcam, RRID:AB_880494) anti-PRPH (1:100; AB1530, Millipore, RRID:AB_90725), anti-CDX2 (1:200; ab76541, Abcam, RRID:AB_1523334), anti-ASCL1 (1:100; 556604, BD Pharmigen, RRID:AB_396479), anti-GATA3 (1:100; sc-269, Santa Cruz, RRID:AB_627666), anti-GFP (1:1000; ab13970, Abcam, RRID:AB_300798), anti-ISL1 (1:100, DSHB), anti-PHOX2B (1:100; sc-376997, Santa Cruz), anti-ETS1 (1:200; D8O8A, CST), anti-NANOG (1:500; 1E6C4 CST, RRID:AB_10548762), anti-OTX2 (1:40; AF1979, R and D Systems, RRID:AB_2157172), anti-DBH (1:250; AB1585, Millipore, RRID:AB_90805), anti-TUBB3 (1:1000, ab18207, Abcam, RRID:AB_2256751). Images were processed using Photoshop and Fiji. Nuclear segmentation followed by single cell fluorescence quantification was performed either using Fiji (RRID:SCR_002285) and the MultiCell3D application as described previously (*Tsakiridis et al., 2014*; *Osorno et al., 2012*) or CellProfiler (*Carpenter et al., 2006*) (RRID:SCR_007358) using a custom made pipeline. Cells stained with secondary antibody only were used as a negative control to set a threshold fluorescence intensity. Following nuclear segmentation, the fluorescence intensity of each channel was masked back to nuclei and gave the number of positive (with fluorescence intensity greater than secondary only control) and negative cells per channel.

## Electrophysiology

Whole-cell patch clamp was used to record membrane currents or membrane potentials from single hPSC-derived sympathetic neurons (n = 14), at room temperature (20–25°C), using an Optopatch (Cairn Research Ltd, UK) patch clamp amplifier. The extracellular solution contained (mM): 135 NaCl, 5.8 KCl, 1.3 CaCl2, 0.9 MgCl2, 0.7 NaH2PO4, 5.6 D-glucose, 10 Hepes-NaOH, 2 sodium pyruvate. Amino acids and vitamins for Eagle's minimal essential medium (MEM) were added from concentrates (Invitrogen, UK). The pH was adjusted to 7.5 and the osmolality was about 308 mosmol kg-1. hPSC-derived sympathetic neurons were viewed using an upright microscope equipped with Nomarski DIC optics (Nikon, Japan) and were continuously perfused with extracellular solution. Patch electrodes were pulled from soda glass capillaries (Hilgenberg GmbH, Germany) and electrodes had resistances in extracellular solution of around 4 MΩ. The shank of the electrode was coated with surf wax to minimise the fast electrode capacitative transients. The pipette solution contained (mM): 131 KCl, 3 MgCl2, 1 EGTA-KOH, 5 Na2ATP, 5 Hepes-KOH, 10 sodium phosphocreatine (pH 7.3, 290 mosmol kg-1). Voltage and current clamp protocol application and data acquisition were performed using pClamp software and a Digidata 1440A (Molecular Devices, USA). Recordings were filtered at 2.5 or 10 kHz (8-pole Bessel), sampled at 5 or 100 kHz and stored on computer for off-line analysis using Clampfit, GraphPad Prism (GraphPad Software Inc, USA) and Origin (OriginLab, USA) software. Recordings and reported currents were corrected off-line for linear leakage and residual capacitative transients. Membrane potentials under voltage clamp were corrected for the voltage drop across the residual series resistance (Rs) at steady-state current level and for a liquid junction potential, measured between pipette and bath solutions, of –4 mV. Residual Rs after compensation (up to 80%) was 1.2 ± 0.1 MΩ (n = 14) and cell membrane capacitance was 11.4 ± 2.1 pF (n = 14). Statistical comparisons of means were made using either the unpaired student's two-tailed t-test for two data sets, or for comparisons of multiple data sets, using analysis of variance (two-way ANOVA followed by the Bonferroni post-test). For all the above statistical tests $p < 0.05$ was used as the criterion for statistical significance. Mean values are quoted ±S.D. in text and figures.

## Catecholamine ELISA

Commercial ELISA kits were used to assess secretion of Dopamine (Biovision Cat #E4219-100) and Norepinephrine (Biovision cat #4360–100) following the manufacturer's instructions and analysed using a Varioskan Flash plate reader at 450 nm (Thermo Scientific). Sympathetic neurons were depolarised following treatment with 50 mM KCl dissolved in differentiation media. Media supernatant was collected after 10 min and centrifuged at 10000xg for 10 min at 4°C to remove insoluble debris and dead cells and stored at −20°C until analysis. Supernatant from undifferentiated hPSCs was used as a negative control.

## Acknowledgements

We would like to thank Lesley Forrester, Roger Pedersen, Gabsang Lee and Lorenz Studer for providing the SFCi55-ZsGr, T-VENUS, PHOX2B-GFP and SOX10-GFP reporter hPSC lines respectively. Also we are grateful to Merete Long for help with the ELISA experiment. TF and MW are supported by a University of Sheffield, Biomedical Science Departmental PhD studentship. AT is supported by funding from the BBSRC (New Investigator Research Grant, BB/P000444/1), the Royal Society (RG160249) and the Children's Cancer and Leukaemia Group/Little Princess Trust (CCLGA 2016 01). KA and PWA are supported by the EU 7th Framework project PluriMes. MG was supported by a BBSRC grant (BB/J015539/1). JB is supported by the Francis Crick Institute which receives its funding from Cancer Research UK (FC001051), the UK Medical Research Council (FC001051), and the Wellcome Trust (FC001051). VW is supported by an MRC Programme Grant (Mr/K011200/1). MRG is funded by the Italian Ministry of Research (Iteromics Flagship project). SLJ is a Royal Society University Research Fellow. We would like to thank Marta Milo, Vicki Metzis, Celine Souilhol, Ben Steventon, Matt Towers and Heiko Wurdak for advice and critical reading of the manuscript.

## Additional information

### Funding

| Funder | Grant reference number | Author |
|---|---|---|
| University of Sheffield | Biomedical Science Departmental PhD studentship | Matthew Wind |
| Seventh Framework Programme | FP7/2007-2013 agreement no. 602423 (Plurimes) | Konstantinos Anastassiadis Peter W Andrews |
| Biotechnology and Biological Sciences Research Council | BB/J015539/1 | Mina Gouti |
| Cancer Research UK | FC001051 | James Briscoe |
| Wellcome | FC001051 | James Briscoe |
| Medical Research Council | FC001051 | James Briscoe Valerie Wilson |
| Medical Research Council | Mr/K011200/1 | Valerie Wilson |
| Royal Society | University Research Fellow | Stuart L Johnson |
| Ministero dell'Istruzione, dell'Università e della Ricerca | Iteromics Flagship project | Mario R Guarracino |
| Biotechnology and Biological Sciences Research Council | BB/P000444/1 | Anestis Tsakiridis |
| Royal Society | RG160249 | Anestis Tsakiridis |
| Little Princess Trust and Children's Cancer and Leukaemia Group | CCLGA 2016 01 | Anestis Tsakiridis |

The funders had no role in study design, data collection and interpretation, or the decision to submit the work for publication.

## Author contributions

Thomas JR Frith, Conceptualization, Resources, Formal analysis, Investigation, Writing—review and editing; Ilaria Granata, Data curation, Formal analysis, Investigation, Writing—review and editing; Matthew Wind, Erin Stout, Dylan Stavish, James OS Hackland, Stuart L Johnson, Marysia Placzek, Investigation; Oliver Thompson, Paul R Heath, Investigation, Writing—review and editing; Katrin Neumann, Daniel Ortmann, Resources; Konstantinos Anastassiadis, Resources, Writing—review and editing; Mina Gouti, Funding acquisition, Investigation, Writing—review and editing; James Briscoe, Valerie Wilson, Funding acquisition, Writing—review and editing; Mario R Guarracino, Peter W Andrews, Supervision, Funding acquisition, Writing—review and editing; Anestis Tsakiridis, Conceptualization, Resources, Formal analysis, Supervision, Funding acquisition, Investigation, Writing—original draft, Project administration, Writing—review and editing

## Author ORCIDs

Thomas JR Frith https://orcid.org/0000-0002-6078-5466
James OS Hackland https://orcid.org/0000-0001-7087-9995
Konstantinos Anastassiadis http://orcid.org/0000-0002-9814-0559
James Briscoe http://orcid.org/0000-0002-1020-5240
Valerie Wilson http://orcid.org/0000-0003-4182-5159
Anestis Tsakiridis http://orcid.org/0000-0002-2184-2990

## Decision letter and Author response

Decision letter https://doi.org/10.7554/eLife.35786.033
Author response https://doi.org/10.7554/eLife.35786.034

# Additional files

## Supplementary files

• Supplementary file 1. Significantly up- and downregulated transcripts, GO enrichment and TF signatures from RNAseq analysis
DOI: https://doi.org/10.7554/eLife.35786.022

• Supplementary file 2. List of genes upregulated in different NC populations and GO enrichment analysis
DOI: https://doi.org/10.7554/eLife.35786.023

• Supplementary file 3. List of transcription factors shared between different NC populations
DOI: https://doi.org/10.7554/eLife.35786.024

• Supplementary file 4. List of all genes up- and down-regulated in indicated NC populations and their progenitors.
DOI: https://doi.org/10.7554/eLife.35786.025

• Supplementary file 5. List of primers
DOI: https://doi.org/10.7554/eLife.35786.026

• Transparent reporting form
DOI: https://doi.org/10.7554/eLife.35786.027

## Data availability

The microarray and RNAseq data have been deposited to GEO (GSE109267 and GSE110608).

The following datasets were generated:

| Author(s) | Year | Dataset title | Dataset URL | Database, license, and accessibility information |
|---|---|---|---|---|
| Heath PR | 2018 | Axial progenitors generate trunk neural crest cells at a high efficiency in vitro | https://www.ncbi.nlm.nih.gov/geo/query/acc.cgi?acc=GSE109267 | Publicly available at the NCBI Gene Expression Omnibus (accession no: GSE109267) |

| Granata I, Tsakiridis A | 2018 | RNA sequencing analysis of human embryonic stem cells and axial progenitors | https://www.ncbi.nlm.nih.gov/geo/query/acc.cgi?acc=GSE110608 | Publicly available at the NCBI Gene Expression Omnibus (accession no: GSE110608) |

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
