## [Decision Letter]

[Editors’ note: the authors were asked to provide a plan for revisions before the editors issued a final decision. What follows is the editors’ letter requesting such plan.]

Thank you for sending your article entitled "Human axial progenitors generate trunk neural crest cells" for peer review at *eLife*. Your article is being evaluated by three peer reviewers, and the evaluation is being overseen by a Reviewing Editor and Sean Morrison as the Senior Editor.

As you will see from the reviewer comments below, there was enthusiasm for the technique, but there were two significant issues that preclude us from making a decision.

A) There was a concern about the quantification of the day 3 NMP cell fates. Based on the data, there is a significant possibility that there are pluripotent stem cells present at day 3, and the reviewers feel that a more robust quantification of these cell fates is necessary. Although this could be done using FACS based approaches, as an alternative, a high content imaging-based quantitation could be used, assuming there is not too much clustering of the cells in 3D. Carefully tracking cells at day 3 is important, as some of the pitfalls may include the possibility of pluripotent stem cells left at day 3 (e.g. Figure 4G showing pluripotent markers still enriched in NMP stage) which could change the interpretation of the study.

B) An in vivo functional readout of the derived cells is required to establish the true importance of the technique. For example, you could transplant the cells into chick embryos to test their capacity to form neurons that express sympathetic markers in sympathetic ganglia, as has been done in prior studies (as in Figure 6 of Denham et al.et al., 2015).

We recognize that these experiments can be complex and time-consuming, and *eLife* generally asks that all revision experiments be performed within 2 months. Given this, we would like you to examine the list of all reviewer comments below as well as the comments discussed above, and decide if you think this is feasible within an approximately 2 month period. If so, then the editors and reviewers invite you to respond within the next two weeks with a detailed action plan and timetable for the completion of the additional work. We plan to share your responses with the reviewers and then issue a decision.

*Reviewer #1:*

The study by Tsakiridis and colleagues combines the induction of a posterior neuromesodermal progenitor (NMP) fate with the induction of a posterior (trunk) neural crest (NC) identity and the derivation of trunk NC-derived sympathoadrenal progenitors (SAPs). Overall, the study is well designed and clearly written. However, protocols to trigger NMPs from mouse and human PSCs are already very well established across the field. Furthermore, two groups (as cited by the authors in the current manuscript: Friedel et al., 2015; Guimond et al., 2014) have already provided some evidence of NMP-derived neural crest derivatives, in particular SAP derived chromaffin-like cells. Nevertheless, the current study goes beyond those published studies by focusing more on developmental approaches and by looking in more detail at the transitions from NMP induction to NC specification. In particular, the study presents detailed analyses of HOX gene induction and global gene expression, and it achieves a protocol to generate trunk NC and SAP induction at high efficiencies.

Beyond the potential concerns about novelty mentioned above, there are additional issues that should be addressed including better quantification of the various subpopulations observed and the need to more clearly demonstrate the lineage relationship of NMP and subsequent NC identity. Furthermore, the characterization of the resulting SAP-like neural crest derivatives is quite limited and the trunk identity of the resulting neurons highly dependent on a few reagents such as HOXC9 antibodies which are not validated very well in the current study.

1) IHC vs Flow based quantification. The study relies very heavily on the use of immunocytochemical (ICC) analysis including double and triple stains. Such analysis can be tricky as it does not allow for an easy representation of the different levels of expression among positive cells and, for many of the Figures presented here, it lacks quantification. This issue could be addressed by using intracellular flow, where both quantification and representation of levels across cells can be shown more easily. For example, it would be important to have proper quantification and presentation of expression levels for the various panels in Figure 1 as well as Figure 2B among others.

2) Lack of a true lineage analysis. The authors argue lineage-relationships based the co-expression of certain markers rather than any true genetic lineage mapping or at least cell sorting based analyses. The study could be much more convincing by including genetic fate mapping data, ideally using a hPSC reporter line such as a reporter line for T. At the minimum, the authors should present a more detailed quantification of marker expression as detailed under #1 and discuss potential pitfalls of their current approach (such as the possibility of persisting OCT4+ cells at NMP stage – see #4 below – or early NC commitment in parallel or prior to NMP stage).

3) BMP/LDN experiments. The authors present data to show that endogenous BMP signaling seems to be important for the expression of early NC markers in day 3 NMP cultures. Indeed, LDN treatment greatly reduces several of those markers. However, this gets back to the question of lineage relationships and whether NMPs indeed are fully plastic as suggested or whether a subset is already committee to NC lineage and therefore no longer bona fide NMPs. Another interesting point regarding those data is that LDN also reduces CDX2 levels. This argues that BMP may not only trigger NC bias but also enable CDX2 expression and posterior patterning. Do the authors have data on LDN treated NMPs that were subsequently induced towards NC lineage by using the BMP/WNT NC protocol from day 3 onwards? Do they observe comparable potential to trigger NC after early LDN treatment?

4) Persisting OCT4 expression the gene expression data in Figure 4G shows highly differential expression of OCT4 in NMP vs NMP-NC (~ 1,000 fold differential expression) – this suggests that there is remaining OCT4 expression either in all NMPs, or there is a subset of cells at NMP stage that retain OCT4 expression. This should be addressed by ICC or flow cytometry for OCT4. If there is a subset of OCT4+ cells at day 3, it raises concerns along the lines stated under #1 and #2 regarding lineage origin of those NC cells.

5) Subtype identity of NMPs There are conceptual questions about how to define NMPs if the cells already express NC (and potentially other subtype) markers at the day 3 stage prior to NC induction. It raises the issue of how many NMP subpopulations co-exist and where did they come from (a defined, common intermediate?). For example, the authors state: "Collectively, these findings indicate that a NC/border state probably arises within multipotent posterior axial progenitors which have not committed to a neural or mesodermal fate" However, if cells already express PAX3 and SOX9, it seems tricky to argue that those cells within NMP cultures are not committed at all.

6) HOXC9. The HOXC9 antibody is a critical reagent for this study. The induction of HOXC9 levels seem to vary quite a bit based on the gene expression data presented throughout the paper while the antibody seems to label a very large proportion of all the cells. It would be helpful to include some additional control assays to demonstrate the specificity of the antibody, and to include stage-matched ANCs or RA treated ANCs throughout to further validate specificity.

7) HOX co-expression data. There are no data looking at co-expression of various HOX genes. This is particularly relevant as the authors conclude that NMP-derived NC is HOX6-9 while ANC is HOX1-5 (e.g. Summary scheme in Figure 6). However, the gene expression data indicate that most anterior HOX genes (1-5) are expressed equally or even higher in NMP vs ANC or ANC+RA derived lineages (e.g. Figure 1C and Figure 3E, Figure 4D).

8) Characterization of trunk NC derivatives / SAPs A key argument regarding the importance of the current study, is the ability to efficiently generate trunk NC derivatives such as sympathetic neurons without the need for sorting. However, the characterization of those sympathetic neurons is very limited and lacks basic co-expression data (e.g. TH/DBH) or functional data such as evidence of catecholamine release or other physiological hallmarks of sympathetic neurons. Such assays are commonly presented by other studies reporting on the derivation of sympathetic neurons from hPSCs (e.g. Friauf, 2008).

*Reviewer #2:*

The study by Frith and colleagues employs methods to convert human pluripotent stem cells (hPSC) into posterior/caudal neural crest (NC) cells via an axial progenitor intermediate. In addition, the authors define culturing conditions that allow efficient production of peripheral sympathetic neurons without the need for additional purification schemes, such as FACS. Overall, the authors present their work very well, with excellent command of the literature, and have advanced our knowledge for generating trunk NC cells and their derivatives that I expect will be used by researchers in the field. However, in its current form, the conclusions are over-stated and the impact of the study is lessened by the fact that similar protocols for generating trunk NC and peripheral neurons (and other NC-cell types) have recently been published (as noted by the authors). In addition, the experimental approaches are not overly innovative (cell culture, immunohistochemistry and RNA-Seq) and largely focus on sufficiency outcomes with no experiments addressing necessity. As one of the main conclusions from the study concerns the origin of trunk NC cells from axial progenitors, there should be some spatial data to confirm the molecular signatures. For example, transplantation of anterior versus posterior NC cells into avian embryos to show the observed transcriptional changes impact the fate of cells at different axial positions. This technique is commonly used in the field to test axial identity and NC competence (e.g., Abu-Bonsrah et al., 2017, Denhamn et al., 2017) and would complement the heavy reliance on using only molecular signatures in cell culture. Indeed, many of the genes used by the authors to define axial NC identity are expressed in other cell types, particularly other neural, neuroectodermal and mesoderm derivatives that develop adjacent to NC cell in vivo. Thus, without the addition of fate mapping data in vivo, other interpretations/conclusions could be drawn on what cell types are actually generated by this new method..

1) The authors should employ transplantation techniques with different progenitors derived in vitro (axial, NMPs, PXM, and anterior versus posterior NC cell types) to show they indeed are fated/committed to specific axial lineages in vivo. For example, RA-treated anterior NC cells should largely generate cardiac/vagal cell types in vivo compared to posterior NC cells, whereas transplanted axial cells would be expected to generate multiple posterior lineages. Such data would significant enhance the impact of the study and strengthen the conclusion that human axial progenitors generate trunk NC.

2) The authors identify a potential new gene involved in trunk NC development, ASLX3. The authors should perform in situ hybridization or immunohistochemistry on vertebrate animals (mouse, chick or fish) to localize ASLX3 (or its conserved equivalent) to trunk NC cells in vivo. This would strengthen their argument that their approach used to classify axial identity in vitro (which is reliant on HOX gene expression) can also produce new knowledge on the genetic regulation of trunk NC in vivo.

*Reviewer #3:*

The manuscript shows that trunk-NC cells are best/most efficiently derived via an axial progenitor population, similar to the neuro-mesodermal progenitor giving rise to lower motor neurons. The authors show solid work that clearly indicate that this progenitor can be guided into producing NC cells with trunk identity. More sophisticated experiments, such as following cells over time using reporter lines to show that the NMP truly is giving rise to the trunk-NC population as well as in vivo work showing the same thing would definitively prove this interesting hypothesis. Therefore, some of the conclusions should be worded a bit more carefully. Nevertheless, this is the first report (to my knowledge) of NC cells derived from NMPs with trunk NC character and it seems a highly efficient one as well. Thus, I think this is an important report that should be published.

- To strengthen the conclusion in subsection “A BMP-dependent neural crest signature in human axial progenitor cultures” (Collectively, these findings indicate that a NC/border state probably arises within multipotent posterior axial progenitors, which have not committed to a neural or mesodermal fate) it would be useful to see what percentage of all cells are posterior axial progenitors committed to become trunk-NC. For example, by doing triple stained intracellular FACS analysis of T, CDX, SOX9.

- In Figure 3D, the authors show that the axial progenitor can give rise to paraxial mesoderm. This could be strengthened by showing that these PXM cells can further be differentiated into definitive mesodermal cell types.

- Do the SOX9+ cells early at day 3 also express SOX10? (associated with Figure 2B). Why does SOX9 come on before SOX10 and does SOX10 stay on?

- Can you maintain this trunk-NC cells as progenitors and expand them in culture, if that is possible it would make this technique even more useful.

-The RNA-seq/microarray data suggests that DMBX1 and LHX5 versus HES6 could serve as markers to distinguish anterior from trunk NC cells. This notion should be strengthened by staining to show the same at the protein level.

- Figure 5C: 80% of Phox2B-GFP- cells are ASCL1+? What are these cells? Maybe they are younger cells that have not yet turned on Phox2B, which would mean that at day 18 there is a mixture of cells in terms of developmental stage. Could that be showed more?

- Scale bars are missing in all pictures.

- Discussion section: This should be stated more carefully, since the transcriptome analysis is done in in vitro generated cells. Without confirming in vivo data, one cannot definitively prove that axial progenitors are converted into posterior NC in embryogenesis.

---

## [Author Response]

[Editors' note: the authors’ plan for revisions was approved and the authors made a formal revised submission.]

As you will see from the reviewer comments below, there was enthusiasm for the technique, but there were two significant issues that preclude us from making a decision.A) There was a concern about the quantification of the day 3 NMP cell fates. Based on the data, there is a significant possibility that there are pluripotent stem cells present at day 3, and the reviewers feel that a more robust quantification of these cell fates is necessary. Although this could be done using FACS based approaches, as an alternative, a high content imaging-based quantitation could be used, assuming there is not too much clustering of the cells in 3D. Carefully tracking cells at day 3 is important, as some of the pitfalls may include the possibility of pluripotent stem cells left at day 3 (e.g. Figure 4G showing pluripotent markers still enriched in NMP stage) which could change the interpretation of the study.

The presence of OCT4 in hPSC-derived axial progenitors/NMPs is not unexpected as it has been shown to be expressed within non-pluripotent cells in the posterior of the early somite mouse embryo including NMPs (Yeom et al., 1996; Osorno, Tsakiridis et al., 2012; Aires et al., 2016; Gouti et al., 2017) and has a role in the regulation of axis elongation (Aires et al., 2016; DeVeale et al., 2013). Thus, to investigate the persistence of pluripotent cells, we examined the presence of cells expressing more definitive pluripotency markers such as NANOG (Osorno, Tsakiridis et al., 2012) and OTX2 (Acampora et al., 2013) in axial progenitor cultures derived from a T-VENUS reporter hPSC line (Mendjan et al., 2014). Following quantification, we found a small (5% of total cells) NANOG+T-VENUS+^low^ fraction and no OTX2+T-VENUS+ co-expressing cells. This indicates minimal contamination of our NC-potent T+ axial progenitors with pluripotent cells. We have included these data in new Figure 3—figure supplement 3. However, to eliminate completely the possibility that pluripotent cells may be the source of trunk NC cells following re-plating of axial progenitor cells in NC-inducing conditions we also flow- sorted and tested the NC potential of T-VENUS+^high^ cells which we found to be NANOG negative (see new Figure 3—figure supplement 3C). Our new results show that indeed these cells exhibit the potential to generate SOX10+ neural crest cells, a proportion of which (approximately one third) is also positive for HOXC9 (See new Figure 3F). This implies that T+^high^ axial progenitors are the source of at least half of the trunk HOXC9+SOX10+ NC cells we obtain following differentiation of bulk unsorted human axial progenitor cultures (about 50% of the total cells, see Figure 3C). We speculate that the rest is likely to arise from the T+^low^ fraction, which was not included in the sorted population. These new data also suggest that the production of trunk NC from T+ progenitors does not occur via “posteriorisation” of an OTX2+ anterior NC progenitor entity (Simoes-Costa and Bronner, 2016) since the very few OTX2+ cells we detected in our axial progenitor cultures were all T-VENUS negative and therefore were not included in the sorted population that gave rise to SOX10+HOXC9+ cells upon plating in NC-inducing conditions (Figure 3F, Figure 3—figure supplement 3B, D in revised m/s). We have discussed these findings in the relevant Results section (“To further confirm the lineage relationship between trunk NC cells and T+ axial progenitors… originate from T-expressing cells” and “This is further supported by our T-VENUS… “caudalisation” of an anterior OTX2+ NC precursor”).

B) An in vivo functional readout of the derived cells is required to establish the true importance of the technique. For example, you could transplant the cells into chick embryos to test their capacity to form neurons that express sympathetic markers in sympathetic ganglia, as has been done in prior studies (as in Figure 6 of Denham et al.et al., 2015).

We have now verified the functionality of the axial progenitor-derived trunk NC cells using the following assays:

(i) We show that trunk NC cells differentiated from a fluorescent reporter human iPSC line available in our lab (Lopez-Yrigoyen et al., 2018) and transplanted into/on top of the dorsal neural tube of chick embryos behave similarly to embryonic NC by exhibiting migratory behaviour (6/6 grafted embryos) and colonising the DRG where they were found to upregulate markers indicative of their location (3/6 grafted embryos). These data are shown in new Figure 3D and Figure 3—figure supplement 2A-C.

(ii) We show by ELISA that axial progenitor-derived sympathetic neurons produce catecholamines (dopamine and norepinephrine) at levels similar or higher to the ones described previously for in vitro derived sympathetic neurons (Oh et al., 2016). These data are shown in new Figure 5G.

(iii) We confirm that axial progenitor-derived sympathetic neurons exhibit an action potential firing profile similar to the one described previously for in vitro derived sympathetic neurons (Oh et al., 2016). These data are shown in new Figure 5F and Figure 5—figure supplement 1D.

We recognize that these experiments can be complex and time-consuming, and eLife generally asks that all revision experiments be performed within 2 months. Given this, we would like you to examine the list of all reviewer comments below as well as the comments discussed above, and decide if you think this is feasible within an approximately 2 month period. If so, then the editors and reviewers invite you to respond within the next two weeks with a detailed action plan and timetable for the completion of the additional work. We plan to share your responses with the reviewers and then issue a decision.Reviewer #1:The study by Tsakiridis and colleagues combines the induction of a posterior neuromesodermal progenitor (NMP) fate with the induction of a posterior (trunk) neural crest (NC) identity and the derivation of trunk NC-derived sympathoadrenal progenitors (SAPs). Overall, the study is well designed and clearly written. However, protocols to trigger NMPs from mouse and human PSCs are already very well established across the field. Furthermore, two groups (as cited by the authors in the current manuscript: Friedel et al., 2015; Guimond et al., 2014) have already provided some evidence of NMP-derived neural crest derivatives, in particular SAP derived chromaffin-like cells. Nevertheless, the current study goes beyond those published studies by focusing more on developmental approaches and by looking in more detail at the transitions from NMP induction to NC specification. In particular, the study presents detailed analyses of HOX gene induction and global gene expression, and it achieves a protocol to generate trunk NC and SAP induction at high efficiencies.Beyond the potential concerns about novelty mentioned above, there are additional issues that should be addressed including better quantification of the various subpopulations observed and the need to more clearly demonstrate the lineage relationship of NMP and subsequent NC identity.

We have included new quantification data showing the correlation between T (=NMP/axial progenitor marker) and NC/border markers such as SOX9, SNAI2 and PAX3 using two different hPSC cell lines (MasterShef7 and the H9-derived T-VENUS line)- see data in new Figure 2C and Figure 2—figure supplement 1B.

Furthermore, the characterization of the resulting SAP-like neural crest derivatives is quite limited and the trunk identity of the resulting neurons highly dependent on a few reagents such as HOXC9 antibodies which are not validated very well in the current study.

We have now included a number of additional functional experiments to address this issue, see our response above to point B.

1) IHC vs Flow based quantification. The study relies very heavily on the use of immunocytochemical (ICC) analysis including double and triple stains. Such analysis can be tricky as it does not allow for an easy representation of the different levels of expression among positive cells and, for many of the Figures presented here, it lacks quantification. This issue could be addressed by using intracellular flow, where both quantification and representation of levels across cells can be shown more easily. For example, it would be important to have proper quantification and presentation of expression levels for the various panels in Figure 1 as well as Figure 2B among others.

We have carried out a thorough quantification of the expression of different NMP/NC/border/pluripotency markers in axial progenitor cultures following immunostaining and image analysis since in our hands intracellular FACS often results in high background and relies heavily on the quality of the antibodies used. These results are now part of figure 2C, Figure 2—figure supplement 1B, Figure 3—figure supplement 3A, B; see also our responses above.

2) Lack of a true lineage analysis. The authors argue lineage-relationships based the co-expression of certain markers rather than any true genetic lineage mapping or at least cell sorting based analyses. The study could be much more convincing by including genetic fate mapping data, ideally using a hPSC reporter line such as a reporter line for T. At the minimum, the authors should present a more detailed quantification of marker expression as detailed under #1 and discuss potential pitfalls of their current approach (such as the possibility of persisting OCT4+ cells at NMP stage – see #4 below – or early NC commitment in parallel or prior to NMP stage).

See our response, above, to point A.

3) BMP/LDN experiments. The authors present data to show that endogenous BMP signaling seems to be important for the expression of early NC markers in day 3 NMP cultures. Indeed, LDN treatment greatly reduces several of those markers. However, this gets back to the question of lineage relationships and whether NMPs indeed are fully plastic as suggested or whether a subset is already committee to NC lineage and therefore no longer bona fide NMPs. Another interesting point regarding those data is that LDN also reduces CDX2 levels. This argues that BMP may not only trigger NC bias but also enable CDX2 expression and posterior patterning. Do the authors have data on LDN treated NMPs that were subsequently induced towards NC lineage by using the BMP/WNT NC protocol from day 3 onwards? Do they observe comparable potential to trigger NC after early LDN treatment?

The reviewer is raising some very interesting points. The reduction in *CDX2* transcript levels following BMP inhibition during the induction of NMPs from hPSCs does not appear to be statistically significant (Figure 2E) and we also found that the total number of CDX2 protein positive cells is not affected by LDN treatment (data not shown). Furthermore, our new data show that BMP-inhibited axial progenitors can still generate trunk NC at a similar efficiency to untreated controls (See new Figure 3—figure supplement 2D). These data indicate that endogenous BMP signalling alone is not the mediator of NC potential in axial progenitor cultures and we have therefore amended our statements in the new manuscript version regarding the role of this pathway.

4) Persisting OCT4 expression the gene expression data in Figure 4G shows highly differential expression of OCT4 in NMP vs NMP-NC (~ 1,000 fold differential expression) – this suggests that there is remaining OCT4 expression either in all NMPs, or there is a subset of cells at NMP stage that retain OCT4 expression. This should be addressed by ICC or flow cytometry for OCT4. If there is a subset of OCT4+ cells at day 3, it raises concerns along the lines stated under #1 and #2 regarding lineage origin of those NC cells.

See our response to point A, above.

5) Subtype identity of NMPs There are conceptual questions about how to define NMPs if the cells already express NC (and potentially other subtype) markers at the day 3 stage prior to NC induction. It raises the issue of how many NMP subpopulations co-exist and where did they come from (a defined, common intermediate?). For example, the authors state: "Collectively, these findings indicate that a NC/border state probably arises within multipotent posterior axial progenitors which have not committed to a neural or mesodermal fate". However, if cells already express PAX3 and SOX9, it seems tricky to argue that those cells within NMP cultures are not committed at all.

The question of NMP subpopulations is very interesting and requires further investigation but it is outside the scope of this manuscript. The lack of an exclusive NMP marker makes the precise definition of committed vs non-committed NMPs very challenging. Recent lineage tracing and single cell RNA-seq studies have revealed that mouse embryonic NMP subpopulations exhibit expression of more definitive lineage-specific markers such as the paraxial mesoderm gene Tbx6 (Javali et al., 2017; Gouti et al., 2017). We speculate that co-expression of NC/border markers with NMP markers such as T in day 3 FGF-CHIR treated hPSC cultures probably represents a similar phenomenon. Our new T-VENUS-sorting experiments demonstrating that at least some of the T+ axial progenitors (which are presumably N-M potent since about 70-80% T+ cells are also Sox2+CDX2+, and also co-express border/NC markers, see new Figure 2C, Figure 2—figure supplement 1B) can generate trunk NC cells supports the notion that “a NC/border state probably arises within multipotent posterior axial progenitors which have not committed to a neural or mesodermal fate”.

6) HOXC9. The HOXC9 antibody is a critical reagent for this study. The induction of HOXC9 levels seem to vary quite a bit based on the gene expression data presented throughout the paper while the antibody seems to label a very large proportion of all the cells. It would be helpful to include some additional control assays to demonstrate the specificity of the antibody, and to include stage-matched ANCs or RA treated ANCs throughout to further validate specificity.

We are confident about the specificity of the monoclonal anti-HOXC9 antibody we employ. We did not obtain any positive signal when we used the antibody on undifferentiated hES cells suggesting that it is specific. We also validated the antibody’s specificity by carrying out immunostaining on ETS1+ anterior NC cultures and found no HOXC9 positivity, also in line with our transcriptome analysis data. We have included these data in new Figure 3—figure supplement 1A, B.

7) HOX co-expression data. There are no data looking at co-expression of various HOX genes. This is particularly relevant as the authors conclude that NMP-derived NC is HOX6-9 while ANC is HOX1-5 (e.g. Summary scheme in Figure 6). However, the gene expression data indicate that most anterior HOX genes (1-5) are expressed equally or even higher in NMP vs ANC or ANC+RA derived lineages (e.g. Figure 1C and Figure 3E, Figure 4D).

The reviewer is raising a valid point. We should note that co-expression of HOX1-5 and HOX6-9 transcripts does occur in the posterior part of the mouse embryo around E9.5, both in the NMP-containing tailbud region (Gouti et al., 2017l) as well as in differentiated posterior neural/neural crest cells (e.g. for *Hoxb1* expression in E9.5 mouse embryos see Glaser et al. Development 2006 and Arenkiel et al., 2003; for *Hoxc9* expression see Bel et al., 1998) and thus the presence of HOX1-5 transcripts in HOX9+ trunk NC cells is not unexpected. Alternatively, the co-expression of transcripts belonging to both HOX1-5 and HOX6-9 groups may be a result of the co-emergence of a separate population of cardiac/vagal NC cells during trunk NC differentiation. This may be due to the action of endogenous RA signalling since our microarray data revealed the upregulation of RA signalling components in axial progenitor-derived trunk NC even though no exogenous retinoic acid was added to the differentiation medium (see Supplementary file 2 in submitted manuscript). Unfortunately, we have not found reliable antibodies against HOX group 1-5 members that we can use together with HOXC9 in order to distinguish between these two possibilities. We have included a discussion of this point in the Results section in the revised manuscript (“The simultaneous presence… even though no exogenous RA was added to the differentiation medium”).

8) Characterization of trunk NC derivatives / SAPs A key argument regarding the importance of the current study, is the ability to efficiently generate trunk NC derivatives such as sympathetic neurons without the need for sorting. However, the characterization of those sympathetic neurons is very limited and lacks basic co-expression data (e.g. TH/DBH) or functional data such as evidence of catecholamine release or other physiological hallmarks of sympathetic neurons. Such assays are commonly presented by other studies reporting on the derivation of sympathetic neurons from hPSCs (e.g. Friauf, 2008).

See our response to point B. We have now also included data showing co-expression of TH and DBH in axial progenitor-derived sympathetic neurons (see new Figure 5—figure supplement 1B).

Reviewer #2:The study by Frith and colleagues employs methods to convert human pluripotent stem cells (hPSC) into posterior/caudal neural crest (NC) cells via an axial progenitor intermediate. In addition, the authors define culturing conditions that allow efficient production of peripheral sympathetic neurons without the need for additional purification schemes, such as FACS. Overall, the authors present their work very well, with excellent command of the literature, and have advanced our knowledge for generating trunk NC cells and their derivatives that I expect will be used by researchers in the field.

We would like to thank the reviewer for their positive comments.

However, in its current form, the conclusions are over-stated and the impact of the study is lessened by the fact that similar protocols for generating trunk NC and peripheral neurons (and other NC-cell types) have recently been published (as noted by the authors).

We have tried to be careful about our conclusions and we are happy to consider rephrasing them on a case-to-case basis if required. We disagree that the impact of our study is lessened by the publication of similar protocols (we presume the reviewer refers to the manuscripts by Denham et al., 2015 and Abu-Bonsrah et al., 2017) and we would like to point to reviewer 3’s comment that “this is an important report that should be published”. Apart from establishing a highly efficient method for generating trunk NC cells we believe that our manuscript also provides an insight into the basis of anterior-posterior regionalisation in the human neural crest and therefore, as reviewer 1 correctly noted, goes beyond the Denham et al., and Abu-Bonsrah et al., publications, which focus almost exclusively on improving the technical aspects of human ES cell differentiation toward posterior neural crest and its derivatives.

In addition, the experimental approaches are not overly innovative (cell culture, immunohistochemistry and RNA-Seq) and largely focus on sufficiency outcomes with no experiments addressing necessity.

We do not understand this point. We believe that the additional data we have included (e.g. trunk NC cell grafting, catecholamine release assay, electrophysiology, NMP/NC/pluripotency marker quantification, T-VENUS sorting experiment etc.) provide better insight into the functionality and nature of the cell populations we are generating in vitro.

As one of the main conclusions from the study concerns the origin of trunk NC cells from axial progenitors, there should be some spatial data to confirm the molecular signatures. For example, transplantation of anterior versus posterior NC cells into avian embryos to show the observed transcriptional changes impact the fate of cells at different axial positions. This technique is commonly used in the field to test axial identity and NC competence (e.g., Abu-Bonsrah et al.et al., 2017, Denhamn et al.et al., 2017) and would complement the heavy reliance on using only molecular signatures in cell culture.

In our opinion, the use of chick embryo transplantation as a comparative assay for quantifying differences in the axial identity status of heterogeneous hPSC-derived donor cell populations is very challenging to set up and possibly hard to interpret. We predict that the location of the graft site will routinely promote the selection of “compatible” subpopulations which in some cases will be minor “contaminants” co-emerging with the desired cell type during hPSC differentiation (e.g. vagal NC cells arising in trunk NC cultures and vice versa) and thus a large number of grafts will be required in order to derive statistically significant data. Further complications may arise due to the differential effect that the axial identity of the donor cells may exert on their developmental potency/proliferation in a heterotopic grafting context (Zhang et al., 2010). In fact, we are not aware of any study where chick embryo grafting was employed to compare in a quantitative manner distinct PSC-derived cell types in terms of axial identity. However, as stated earlier we have performed grafting of axial progenitor-derived trunk NC cells into chick embryos to confirm in qualitative terms that these behave similarly to their in vivo counterparts, i.e. they exhibit NC-like migratory behaviour and colonise the DRG.

Indeed, many of the genes used by the authors to define axial NC identity are expressed in other cell types, particularly other neural, neuroectodermal and mesoderm derivatives that develop adjacent to NC cell in vivo. Thus, without the addition of fate mapping data in vivo, other interpretations/conclusions could be drawn on what cell types are actually generated by this new method.1) The authors should employ transplantation techniques with different progenitors derived in vitro (axial, NMPs, PXM, and anterior versus posterior NC cell types) to show they indeed are fated/committed to specific axial lineages in vivo. For example, RA-treated anterior NC cells should largely generate cardiac/vagal cell types in vivo compared to posterior NC cells, whereas transplanted axial cells would be expected to generate multiple posterior lineages. Such data would significant enhance the impact of the study and strengthen the conclusion that human axial progenitors generate trunk NC.

See our response above.

2) The authors identify a potential new gene involved in trunk NC development, ASLX3. The authors should perform in situ hybridization or immunohistochemistry on vertebrate animals (mouse, chick or fish) to localize ASLX3 (or its conserved equivalent) to trunk NC cells in vivo. This would strengthen their argument that their approach used to classify axial identity in vitro (which is reliant on HOX gene expression) can also produce new knowledge on the genetic regulation of trunk NC in vivo.

We agree with the reviewer that ASLX3 is a promising candidate marker of trunk NC. However, the characterisation of its expression domains in vivo will be time-consuming since we are not aware of any published antibodies or in situ hybridisation probes that could be used for ASXL3 detection and we would have to generate these reagents from scratch. Consequently, while of clear interest, we think that carrying out these in vivo experiments are beyond the scope of the current study which is focused on the generation of trunk neural crest cells by differentiation of hPSC in vitro.

Reviewer #3:The manuscript shows that trunk-NC cells are best/most efficiently derived via an axial progenitor population, similar to the neuro-mesodermal progenitor giving rise to lower motor neurons. The authors show solid work that clearly indicate that this progenitor can be guided into producing NC cells with trunk identity. More sophisticated experiments, such as following cells over time using reporter lines to show that the NMP truly is giving rise to the trunk-NC population as well as in vivo work showing the same thing would definitively prove this interesting hypothesis. Therefore, some of the conclusions should be worded a bit more carefully. Nevertheless, this is the first report (to my knowledge) of NC cells derived from NMPs with trunk NC character and it seems a highly efficient one as well. Thus, I think this is an important report that should be published.

We would like to thank the reviewer for recognising the importance of our manuscript. We believe that the additional experiments we carried out in response to points A and B (see above) address the reviewer’s suggestions.

- To strengthen the conclusion in subsection “A BMP-dependent neural crest signature in human axial progenitor cultures” (Collectively, these findings indicate that a NC/border state probably arises within multipotent posterior axial progenitors, which have not committed to a neural or mesodermal fate) it would be useful to see what percentage of all cells are posterior axial progenitors committed to become trunk-NC. For example, by doing triple stained intracellular FACS analysis of T, CDX, SOX9.

We have included immunostaining quantification data, see our responses above.

- In Figure 3D, the authors show that the axial progenitor can give rise to paraxial mesoderm. This could be strengthened by showing that these PXM cells can further be differentiated into definitive mesodermal cell types.

We have added data showing the expression of PXM and early somite markers in our axial progenitor-derived PXM cells (see new Figure 3—figure supplement 2E, F).

- Do the SOX9+ cells early at day 3 also express SOX10? (associated with Figure 2B). Why does SOX9 come on before SOX10 and does SOX10 stay on?

We did not detect any SOX10 in day 3 axial progenitors and its expression starts round days 6-7 of differentiation (d0=start of NMP differentiation) peaking round days 8-9 and then becomes downregulated till its extinction round days 10-11. The earlier upregulation of SOX9 relative to SOX10 is in agreement with in vivo expression data (Cheung and Briscoe, 2003).

- Can you maintain this trunk-NC cells as progenitors and expand them in culture, if that is possible it would make this technique even more useful.

We have tried to maintain/expand trunk NC cells as spheres in the same way as we have previously reported for cranial NC (Hackland et al., 2017) but, unfortunately, our attempts have been unsuccessful.

-The RNA-seq/microarray data suggests that DMBX1 and LHX5 versus HES6 could serve as markers to distinguish anterior from trunk NC cells. This notion should be strengthened by staining to show the same at the protein level.

The expression of DMBX1/LHX5 and HES6 in the anterior and trunk NC respectively has already been examined thoroughly in the chick embryo by Simoes-Costa and Bronner, 2016 and therefore we have used them in our study as “established” axial identity readouts.

- Figure 5C: 80% of Phox2B-GFP- cells are ASCL1+? What are these cells? Maybe they are younger cells that have not yet turned on Phox2B, which would mean that at day 18 there is a mixture of cells in terms of developmental stage. Could that be showed more?

We would like to clarify that Figure 5C shows that 80% of the cells were found to be ASCL1+ and approximately 60% were double-positive for both PHOX2B-GFP and ASCL1 and not that 80% of PHOX2B-GFP negative cells are ASCL1+. ASCL1 has been previously shown to be turned on earlier than PHOX2B, both in vivo and in vitro, during hPSC differentiation toward sympathetic neurons (Hirsch et al., 1998; Oh et al., 2016), so we agree with the reviewer that the remaining 20% of ASCL1+ PHXO2B-GFP- cells could be earlier progenitors indicating that day 18 cultures contain cells from different developmental stages. We have found that at later stages almost all PHOX2B-GFP+ cells are also ASCL1+ which further supports the idea that earlier cultures are developmentally heterogeneous. We have made a note of this in the Results section (“At later stages of differentiation almost all PHOX2B-GFP+ cells… more “mature” double positive state”).

- Scale bars are missing in all pictures.

This has been corrected.

- Discussion section: This should be stated more carefully, since the transcriptome analysis is done in in vitro generated cells. Without confirming in vivo data, one cannot definitively prove that axial progenitors are converted into posterior NC in embryogenesis.

We have re-phrased to “Our transcriptome analysis data suggest that, at least in vitro, a generic trunk identity is first installed…”.

Additional references:

Aires R, Jurberg AD, Leal F, Nóvoa, Cohn MJ, Mallo M. Oct4 Is a Key Regulator of Vertebrate Trunk Length Diversity. Dev Cell. 2016; 38(3):262-74.

Cheung M, Briscoe J. Neural crest development is regulated by the transcription factor Sox9. Development. 2003; 130(23):5681-93.

DeVeale B, Brokhman I, Mohseni P, Babak T, Yoon C, Lin A, Onishi K, Tomilin A, Pevny L, Zandstra PW, Nagy A, van der Kooy D. Oct4 is required ~E7.5 for proliferation in the primitive streak. PLoS Genet. 2013; 9(11):e1003957.

Yeom YI, Fuhrmann G, Ovitt CE, Brehm A, Ohbo K, Gross M, Hübner K, Schöler HR. Germline regulatory element of Oct-4 specific for the totipotent cycle of embryonal cells. Development. 1996; 122(3):881-94.

Zhang D, Brinas IM, Binder BJ, Landman KA, Newgreen DF. Neural crest regionalisation for enteric nervous system formation: implications for Hirschsprung's disease and stem cell therapy. Dev Biol. 2010; 339(2):280-94.